# VELOX - A new thermal infrared imager for airborne remote sensing of cloud and surface properties

Michael Schäfer[1], Kevin Wolf[1,*], André Ehrlich[1], Christoph Hallbauer[2], Evelyn Jäkel[1], Friedhelm Jansen[3], Anna Elizabeth Luebke[1], Joshua Müller[1], Jakob Thoböll[1], Timo Röschenthaler[2], Bjorn Stevens[3], and Manfred Wendisch[1]

[1]Leipzig Institute for Meteorology, University of Leipzig, Germany
[2]*enviscope GmbH*, Frankfurt am Main, Germany
[3]Max Planck Institute for Meteorology, Hamburg, Germany
[*]Now at: Institut Pierre-Simon Laplace, Sorbonne Université, Paris, France
**Correspondence:** Michael Schäfer (michael.schaefer@uni-leipzig.de)

**Abstract.** The new airborne thermal infrared (TIR) imager VELOX (Video airbornE Longwave Observations within siX channels) is introduced. VELOX is a commercially available TIR camera system that has been adopted extensively for atmospheric applications, which are introduced in this paper. The system covers six spectral bands with center wavelengths between 7.7 µm and 12 µm. Currently, VELOX is installed on board of the German High Altitude and Long Range Research Aircraft (HALO) to observe cloud and surface properties. It provides observations of two-dimensional fields of upward terrestrial spectral radiance with a horizontal resolution of approximately 10 m by 10 m at a target distance of 10 km. Atmospheric temperature values are rather low compared to the originally intended commercial applications of VELOX and range close to the detection limit of the sensor. This challenge requires additional calibration efforts to enable atmospheric applications of VELOX. Therefore, required sophisticated calibration and correction procedures, including radiometric calibrations, non-uniformity corrections, bad-pixel replacements, and window corrections, are presented. Furthermore, first observations of cloud properties acquired by VELOX during the EUREC[4]A (ElUcidating the RolE of Cloud-Circulation Coupling in ClimAte) campaign are discussed, including an analysis of the cloud top brightness temperature, cloud mask/fraction, and cloud top altitude data. The data reveal the potential of VELOX to resolve the cloud top temperature with a resolution of better than 0.1 K, which translates into a resolution of approximately 40 m in cloud top altitude.

## 1 Introduction

Thermal infrared (TIR) imagery is a well established technique to study atmospheric and surface properties from ground-based, space-based, and airborne platforms. Especially for satellite-based instruments, various retrieval techniques were developed to derive cloud properties like cloud coverage, cloud top temperature and height, thermodynamic phase, and particle effective radius or liquid/ice water path (e.g., Brogniez et al., 2003; Chylek et al., 2006; Iwabuchi et al., 2014, 2016; Platnick, 2017; Someya and Imasu, 2018; Frey et al., 2020) from thermal imagery. Surface property retrievals provide products from thermal

imagery likewise for the surface typing, surface temperature (Haggerty et al., 2003; Kilpatrick et al., 2015; Fu et al., 2020), or snow grain size (Hori et al., 2006).

However, most of the satellite products suffer from a low spatial resolution (often in the range of kilometers), which limits the size of features they can resolve. In the case of clouds, this may lead to uncertainties in the estimation of, e.g., the cloud radiative forcing, if the cloud coverage or the cloud top altitude estimation is wrong (Pavolonis and Key, 2003). For example, the MODerate resolution Imaging Spectrometer (MODIS) on board of Aqua and Terra provides radiance measurements at several TIR channels, but at 1 km spatial resolution only (Esaias et al., 1998; Platnick, 2017). The Advanced Very High Resolution Radiometer (AVHRR; Cracknell, 1997) multi-purpose imaging instrument has three TIR channels, but also at 1 km spatial resolution only. The Visible Infrared Imaging Radiometer Suite (VIIRS; Justice et al., 2013) provides a TIR band for cloud imagery with higher spatial resolution of up to 375 m. Although the Advanced Spaceborne Thermal Emission and Reflection Radiometer (ASTER) on Terra, which has a TIR channel with even higher spatial resolution of 90 m, data is not collected continuously (on average only 8 minutes per orbit; Abrams, 2000). However, because all these sensors are operated on polar orbiting satellites their temporal sampling is limited. Furthermore, they are all probably close to the end of their lifetime. The Multi-Spectral Imager (MSI; Illingworth et al., 2015) instrument on the proposed Earth Clouds, Aerosol and Radiation Explorer satellite mission (EarthCARE; Illingworth et al., 2015) will provide a valuable replacement for, e.g., MODIS. Although it provides fewer channels and covers a smaller swath, the spatial resolution will be increased. However, as a polar orbiting satellite, it will suffer from similar temporal sampling issues. The Geostationary Operational Environmental Satellite (GOES) provides data with one-minute resolution, but due to its geostationary orbit (and hence considerable distance from Earth), the spatial resolution of the TIR channels is only in the range of 2 km (Schmit et al., 2018).

To perform cloud and surface measurements with higher spatial and temporal resolution and to validate satellite products, airborne thermal sensors and imagers were developed. Notable instruments are the enhanced MODIS Airborne Simulator (EMAS-HS; Guerin et al., 2011), or the far infrared radiometer (FIRR; Libois et al., 2016), which helped to develop cloud and surface retrieval products based on TIR observations. Advanced airborne pushbroom TIR imagers are for example the high performance Mineral and Gas Identifier (MAGI; Hall et al., 2015) and MAKO (Hall et al., 2016) and the Airborne Thermal-infrared Hyperspectral Imaging System (ATHIS; Liu et al., 2020), which provide up to 128 spectral channels on up to 2800 spatial pixels.

For the same purpose (increased resolution, satellite validation), airborne imaging spectro-radiometers, radar, and microwave radiometers were developed (e.g., Wirth et al., 2009; Bierwirth et al., 2013; Mech et al., 2014; Ewald et al., 2015; Schnitt et al., 2017; Schäfer et al., 2017) and implemented within the remote sensing package of the High Altitude and LOng Range Research Aircraft (HALO). This sensor package was operated during the Next-generation Aircraft Remote-Sensing for Validation Studies (NARVAL-I, Klepp et al., 2014; Stevens et al., 2019), the NARVAL-II (Stevens et al., 2019), and the North Atlantic Waveguide and Downstream Impact Experiment (NAWDEX, Schäfler et al., 2018) HALO campaigns. The synergy of the HALO remote sensing package was utilized to retrieve various microphysical and macrophysical cloud properties (e.g., De-lanoë et al., 2013; Fricke et al., 2014; Wolf et al., 2019).

With respect to clouds, this suite of instruments provides each a different view on the clouds and different sensitivities to the cloud properties. Based on the single-scattering assumption, active radar (e.g., the HALO Microwave Package, HAMP; Mech et al., 2014; Konow et al., 2019) observations are most sensitive to large cloud droplets and ice crystals and mostly can penetrate the entire cloud. Active lidar (e.g., the Water vapor Lidar Experiment in Space, WALES; Wirth et al., 2009) observations are also sensitive to the backscattering of small cloud droplets and therefore attenuated quickly in liquid clouds.

In comparison, passive instruments are sensitive to a larger range of cloud particle sizes and observe radiation from multiple-scattering within, or emission by, cloud droplets. To interpret and compare solar and thermal observations, their different vertical weightings have to be considered, e.g., the thermal emission originates closer to the cloud top compared to solar radiation, which is to some degree also scattered from lower cloud layers (Platnick, 2000). Therefore, for solar wavelengths, where clouds mostly scatter, the weighting function extends down to the cloud base. In contrast, the emitted thermal radiation

is mostly governed by the properties at the upper most layer (Platnick, 2000). The different vertical weightings highlight the potential of retrieval approaches that combine remotely sensed properties from passive and active instruments operating across a range of frequencies (Schnitt et al., 2017; Wolf et al., 2019; Villanueva et al., 2021).

    For satellite observations, multi-sensor and multi-spectral retrievals are exemplified provided by, e.g., the raDAR/liDAR (DARDAR) cloud classification (Delanoë and Hogan, 2010). These satellite products not only combine radar and lidar obser-

vations, but also consider measurements of the TIR radiation emitted by the clouds at wavelengths between 3.7 μm and 12 μm. Different studies prove the benefit of using TIR radiation for satellite cloud remote sensing applications (e.g., Parol, 1991; Garnier et al., 2012). For example, three-dimensional (3D) radiative effects (shadowing, multiple-scattering, horizontal photon transport), which often cause problems in the solar wavelength range, are avoided, or the detection of clouds above bright surfaces (snow, sea ice) becomes possible due to their different temperatures.

However, for TIR imagers the translation of the raw counts to physical units is challenging. Compared to imagers in the solar spectral range, radiation emitted by the target is not the only source of radiation, which affects the detector signal. The detector itself emits radiation based on its temperature and in general does register a temperature difference between the target and the detector temperature. Therefore, keeping the detector temperature at a known reference is a major requirement, which is either realized by cooling systems or by recording the detector temperature. Furthermore, the radiation reaching the detector

is contaminated by radiation emitted by the imager (body, lens) itself. As it is not always possible that the sensor temperature is stabilized by a cooling system or that the influence of the instruments temperature on the measurements can be well quantified, many TIR imagers apply on-board calibrations with black bodies.

    Here we describe the temperature-stabilized TIR imager VELOX (Video airbornE Longwave Observations within siX channels). VELOX does not apply an on-board calibration, which is compensated by operating the imager in a temperature stabilized

housing and by a series of post-calibration routines. This reduces the size of the imager setup, an advantage that distinguishes VELOX from most of the known airborne TIR imagers. The imaging sensor of VELOX is commercially available and manufactured by the IRCAM GmbH, Erlangen, Germany. It measures radiance in six spectral bands in the TIR wavelength range from 7.7 μm to 12.0 μm, which fits well in the wavelength range that is commonly chosen for TIR measurements and suits common cloud and surface retrieval. Although the number of spectral channels is lower compared to the many of the airborne

TIR imagers introduced above, VELOX provides two-dimensional (2D) images with high temporal resolution. Compared to common line scanners (e.g., MAS, MAGI, or MAKO) this allows a wider range of applications, like analysing the horizontal fine-scale structure of clouds and dynamic processes at clouds edges or stereoscopic image processing utilizing the large overlap of the individual images. VELOX is currently implemented into the remote sensing configuration of HALO (Stevens et al., 2019) for airborne observations, but can potentially be operated on other airborne platforms and as a ground-based sky imager similar to Schäfer et al. (2013) or Jäkel et al. (2013). The integration of VELOX makes the HALO cloud-observatory instrumentation fully analogous to the sensor package being flown on the EarthCARE satellite, combining active radar and lidar observations with passive solar, thermal infrared, and microwave remote sensing.

The VELOX instrumentation and system design for airborne operations is presented in Sect. 2, necessary correction procedures with respect to geometric (pixel size and orientation, image shift due to the use of different filters) and radiometric (radiometric calibration, non-uniform correction, bad-pixel replacement, radiometric cross-calibration, and window correction) calibrations are described in Sect. 3. VELOX was used for the first time in the field during the ElUcidating the RolE of Cloud-Circulation Coupling in ClimAte (EUREC4A; Bony et al., 2017; Konow et al., 2021; Stevens et al., 2021) expedition, which was carried out in January/February 2020 in the vicinity of Barbados. The data collected during the EUREC$^4$A campaign are used in Sect. 4 to demonstrate the capabilities of the new VELOX system. Examples are given for deriving the cloud top brightness temperature, cloud cover, and cloud top altitude.

## 2 Instrumentation

The camera system installed in VELOX comprises two components: an actively cooled TIR imager (VELOX327k veL), and an un-cooled infrared thermometer (Heitronics KT19.85II), which serves as a secondary reference. On board HALO, both instruments are installed in a nadir viewing orientation to observe the upward spectral TIR radiance emitted by clouds and surfaces within the large atmospheric spectral window. The TIR imager has a sensor with 640 by 512 spatial pixels providing 2D images with a maximum frame rate of 100 Hz. The 2D detector is actively temperature controlled and cooled to 65 K by a Stirling cooler. This helps to minimize the influence of the environmental conditions on the performance of the imager, which ensures a wide temperature range for the measurements and a stable absolute calibration. The active cooling reduces the Noise Equivalent Differential Temperature (NEDT) to 40 mK (without filter) and allows for the observation of small temperature differences, in this case of ocean and sea-ice surfaces as well as clouds, with a target temperature between 233.15 K (-40°C) and 373.15 K (100°C). The infrared thermometer is a single sensor radiometer pointing close to the center of the 2D images. The main technical specifications are listed in Tab. 1.

### 2.1 Spectral channel selection

To fulfill the demands of making the TIR imager a tool for cloud and surface characterization, observations at different wavelengths were realized. A synchronously rotating filter wheel, providing six slots for spectral filters (each 25 mm in diameter), is mounted between the lens and the detector. To match images of different channels, when measuring on a fast flying air-

**Table 1.** Specifications of the TIR imager (VELOX327k veL) and the infrared thermometer (KT19.85II).

| Specification | TIR Imager (VELOX327k veL) | Infrared thermometer (KT19.85II) |
|---|---|---|
| Spectral range | 7.7 µm to 12.0 µm | 9.6 µm to 11.5 µm |
| Detector technology | Cadmium-Mercury-Telluride | Thermo-electric |
| Sensor Format | 640 x 512 pixels | 1 pixel |
| Field of view (FOV) | 35.5° x 28.7° | 2.3° |
| Measurable temperature range | -40°C to 100°C | -50°C to 50°C |
| Noise equivalent temperature difference (NETD) | $\leq$ 40 mK, typ. 30 mK @ 25°C | typ. 100 mK |
| Integration time | 5 µs to 150 µs | 5 ms |
| Measurement frequency | 100 Hz | 20 Hz |
| Cooling | active (Sterling) | un-cooled |
| Operation temperature | -15°C to 50°C | -20°C to 70°C |
| Weight | 5 kg | 2.35 kg |
| Power consumption | < 50 W | < 4 W |

craft, the measurement frequency needs to be fast. Therefore, the whole filter wheel rotates with a frequency of 100 Hz. With six channels, this results in a frame rate of about 15 Hz, meaning that a full filter-wheel spin to separately acquire six single images (one with each of the six filters) takes 0.06 sec. Four of the six filter-wheel slots are equipped with band-pass and low-pass filters, which are adapted to the channels (centre wavelength) popular among satellite instruments e.g., as MODIS, MSI, VIIRS, and AVHRR. The applied filter/channel assembly is summarized in Tab. 2, listing their respective spectral range and space-borne sensor equivalents. The mean maximum transmissivity $\mathcal{T}$ of the single filters is in the range between 0.8 and 0.9 (Tab. 2). The smaller the wavelength range of the filter, the smaller the signal reaching the detector, which increases the measurement uncertainty. For some applications, broadband measurements, which provide a higher accuracy are sufficient, e.g., for sea surface temperature or cloud top altitude retrievals. Therefore, two of the filter slots provide redundant broadband measurements (7.7 - 12 µm, Channel 1 and 4 in Tab. 2), but are fitted with an optically transparent window to match the optical paths of the other channels.

Chylek et al. (2006) showed that in the wavelength range between 8.5 µm and 10 µm the imaginary part of the refractive index of liquid water and ice is rather equal, but shows significantly different slopes at wavelengths between 11 µm and 12 µm. Therefore, channels 2 and 4 of the VELOX system are used to discriminate between liquid water and ice in clouds; they can also be used to derive the cloud effective radius or the liquid water path (Iwabuchi et al., 2016; Someya and Imasu, 2018) by applying the split-window method (McMillin, 1980; Brogniez et al., 2003; Heidinger and Pavalonis, 2009).

The two remaining filters (Channel 3 and 6) serve to discriminate ocean, snow, and ice surfaces in Arctic regions. Hori et al. (2006) have shown that the emissivity of snow and ice are rather equal at 10.6 µm, but differ significantly at wavelengths close

**Table 2.** List of all VELOX channels. The wavelength ranges ($\lambda$) of the narrow-band channels are determined by the center wavelength of the particular filter and its half width. The third column lists the mean maximum transmissivity $\mathcal{T}_{\mathrm{max}}$ of the channel filters. According to the wavelength ranges, equivalent MODIS, MSI, VIIRS, and AVHRR bands and center wavelengths are listed.

| Channel | VELOX ($\lambda$) | $\mathcal{T}_{\mathrm{max}}$ | MODIS (Band; $\lambda$) | MSI (Band; $\lambda$) | VIIRS (Band; $\lambda$) | AVHRR (Band; $\lambda$) |
|---|---|---|---|---|---|---|
| 1 | 7.70 - 12.00 µm | 0.94 | - | - | | - |
| 2 | 8.65 ± 0.55 µm | 0.89 | 29; 8.55 µm | 5; 8.80 µm | M14; 8.55 µm | - |
| 3 | 10.74 ± 0.39 µm | 0.83 | 31; 11.03 µm | 6; 10.80 µm | M15; 10.76 µm | 4; 10.8 µm |
| 4 | 7.70 - 12.00 µm | 0.94 | - | - | - | - |
| 5 | 11.66 ± 0.81 µm | 0.88 | 31; 11.03 µm | 7; 12.00 µm | I5; 11.45 µm | - |
| 6 | 12.00 ± 0.50 µm | 0.92 | 32; 12.02 µm | 7; 12.00 µm | M16; 12.01 µm | 5; 12 µm |
| KT19 | 9.60 - 11.50 µm | | | | | |

to 12 µm. Furthermore, those differences at wavelengths close to 12 µm are a function of the particular type of snow and ice (e.g., variable grain size).

The non-imaging nadir pointing infrared thermometer has a smaller FOV and a lower sensitivity than the TIR imager (compare Tab. 1). However, the sensor calibration is more stable, does not need to be cooled and, therefore, is well suited to serve as a reference for the TIR imager. The spectral window of the infrared thermometer covers the spectral channel 3 of the TIR imager (compare Tab. 2), what allows for a measurement comparison between the two instruments and cross-calibration checks.

## 2.2 Airborne operation and installation on HALO

VELOX is currently mounted in the belly pod of HALO, outside the aircraft's pressurized cabin. To operate the sensors under constant environmental conditions and minimize thermal effects on the sensor sensitivity, the TIR imager and the infrared thermometer are installed in a temperature and pressure controlled housing. Humidity inside the housing is minimized by purging the inner air volume with nitrogen gas before each flight. However, the temperature control is only unidirectional, using the instrument's heat production and additional heaters but no Peltier cooling system. To monitor the remaining changes of the environmental conditions inside the chamber, six temperature sensors, an air-pressure sensor, and a humidity sensor are installed, measuring with 1 Hz and relative uncertainties of $\pm\,0.1°C$, $\pm\,2.5\,\mathrm{hPa}$, and $\pm\,2\,\%$, respectively.

Figure 1 shows a technical sketch with an inside view of the single instruments. This cylindrical housing was manufactured mainly out of aluminum and stainless steel and is 44 cm tall with a diameter of 25.5 cm. The entire setup has a weight of approximately 20 kg.

The TIR imager and the infrared thermometer view through the downward looking side of the cylinder, in which transparent windows are integrated. As a trade-off between stability and potential effects on the measurement, two separate Germanium windows of 0.5 cm thickness were chosen. The windows provide a high mean transmissivity $\mathcal{T}$ of 0.9395 for the wavelength

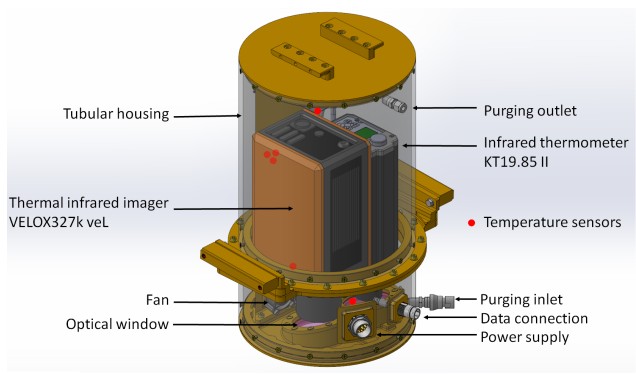

**Figure 1.** Schematic of the cylindrical housing of VELOX. Only the main components of the system are labeled.

range between 7.7 μm and 12 μm. Both sides of the window are covered with an anti-reflection coating to minimize transmission losses and specular reflections inside the housing. In addition, the window plane is tilted by 5° with respect to the TIR imager and infrared thermometer viewing axis. This avoids a direct mirroring of the cold detector itself. The remaining reflections are mostly driven by the thermal emission of the imager optics and are imprinted as a homogeneous offset in the measured radiation field. By knowing the temperature of the instrument's optic and the reflection coefficient of the window, these offsets can be removed (see Sect. 3.2.5).

The window is directly exposed to the air flow around the aircraft and cools down significantly more compared to the atmosphere inside of the VELOX cylinder. Condensation or icing on the inside of the window is avoided by several means. The purging with nitrogen gas reduces the absolute humidity to less than $2\,\mathrm{g\,m^{-3}}$. Furthermore, three heating elements are installed, and three fans generate a continuous airflow over the windows.

## 3 Characterization of the TIR imager

### 3.1 Geometric characterization

The geometric characterization of the VELOX system addresses two main issues: (i) the relative pixel orientation (sensor zenith and azimuth angles) and size (ground or cloud top related) as a function of aircraft attitude and altitude, including the relative position of the field of view (FOV) of the infrared thermometer within the images of the TIR imager, and (ii) the matching procedure of images with different filters.

#### 3.1.1 Viewing geometry

The pixel orientation with respect to the imager's frame (sensor zenith and azimuth angle) is calculated from the detector characteristics (pixel number, detector size) and the imager optics (focal length). The sensor zenith angle is defined as zero in the center of the image and increases towards the sides, reaching $\pm\,17.5°$ in the across-track direction and $\pm\,14°$ in the direction

of flight. During flight, these values need to be adjusted by the pitch and roll angles of the aircraft. The sensor azimuth angle is defined in the mathematically positive direction, with $0°$ in the backward direction. The rectilinear ground pixel size is estimated by trigonometric relations from the sensor zenith angles and the aircraft altitude. In case of clouds and a known cloud top altitude, horizontal pixel size at cloud top can be estimated.

To evaluate the linearity of the pixel orientation, a laboratory calibration following the chessboard approach (compare, Jäkel et al., 2017) was performed by the use of a black body. The calibration showed that the linear calibration of the pixel zenith and azimuth angle is within a $< 1°$ accuracy.

During data processing, the VELOX images are converted from the airborne-fixed coordinate system into an Earth-fixed coordinate system. The attitude angles of HALO were obtained by the aircraft inertial navigation system. Offsets from the
installation of VELOX with respect to the navigation system have been quantified by an inclinometer internally installed in VELOX during test measurements on the ground. During EUREC[4]A, these offsets were below $1°$ and were implemented into the coordinate transformation.

For a target distance of 10 km, the FOV of VELOX translates into an image size of 6.4 km by 5.1 km with an approximated pixel size of 10 m by 10 m. For the same target distance, the infrared thermometer (FOV $= 2.3°$) covers a circular spot with
400 m diameter.

The relative position of the infrared thermometer spot within the images of the TIR imager is identified by calculating the correlation between time series of the infrared thermometer with time series of all TIR imager pixels. The TIR imager pixels with the highest correlations identify the position and size of the infrared thermometer spot in the TIR images, which was found to be shifted by $2.86°$ (approximately 500 m at a distance of 10 km) to the right of the nadir direction with respect to the
flight direction.

### 3.1.2   Image shift due to filter geometry

Although the spectral filters are intended to have a plane surface, even small deviations from this flatness will affect the path of radiation and the focus/mapping of the images on the detector. Therefore, each filter may have a slightly shifted FOV. This needs to be considered during data evaluation, especially when images of different channels are combined. Due to the fixed
position of the filters in the filter wheel, this shift stays fixed as long as the filters are not removed or replaced.

In laboratory measurements, these shifts were quantified to be less than 10 pixels in either direction. However, to provide a consistent data set for all channels, only the image parts that are covered by all channels might be considered in the final data product. After recording a test image (e.g., chess-board like) for each channel, all parts of the scene that are covered by each image are identified manually. Pixels detecting areas outside this identified scene are removed from the image. Afterwards, all
images are slightly smaller, but show perfectly overlapping scenes.

### 3.2   Radiometric calibrations and corrections

Each pixel of the TIR imager provides a digital number, which is proportional to detected photon counts per unit time. During EUREC[4]A, the imager was operated with a frame rate of 100 Hz and an integration time of 70 μs, for which the ther-

mal influence of the spectral filters can be still well corrected for. To convert the digital counts into radiance $I^\uparrow$ (in units of W m$^{-2}$ nm$^{-1}$ sr$^{-1}$) or brightness temperature $T_B$ (here, given in units of °C, while $T_B$ differences are given in units of K), several calibration and correction processes are applied. This includes a radiometric calibration, non-uniform corrections of the detector, extended radiometric cross-calibrations, bad-pixel corrections, and corrections of the influence of the window in the case of airborne application in the tubular mounting. Figure 2 shows an example measurement, acquired during EUREC[4]A. Each step of the calibration and correction process is displayed in a single panel and is separately discussed step by step in the subsequent sections.

### 3.2.1 Internal radiometric calibration model

The TIR imager is provided with an advanced internal radiometric calibration model, developed and validated by the manufacturer over many years (Schreer et al., 2004). This model includes calibration factors to transfer the raw counts into brightness temperature or radiance and takes changes in the imagers body, optics, and filter temperatures into account. This is important, since during research flights those parameters are different from the measurement conditions during the initial radiometric characterization of the imager in the laboratory. Using cross-calibrations in a later step (Sect. 3.2.3), this advanced calibration model is adjusted to the conditions during the research flights.

Figure 2b shows the image after the internal manufacturers radiometric calibration model was applied to the raw image of Fig. 2a. It is obvious that after this processing step, artifacts are still imprinted in the image, e.g., regular vertical stripes, which require further treatment.

### 3.2.2 Non-uniformity correction

TIR imagers with actively cooled detectors typically suffer from the Narcissus effect, which describes the contamination of the image by internal reflections of the sensor on the lens (Lau et al., 1977). Furthermore, each spatial pixel has a slightly different gain and its ground potential has no fixed value. The latter changes slightly after each restart of the imager. Both the different gain and the variable ground potential naturally imprint noise into the measured images. To remove these effects, a two-point non-uniform correction (Budzier and Gerlach, 2015) was applied prior to each flight, which homogenizes the response of all single detector pixels among each other.

The non-uniform correction is performed at the ground and generated after the sensor chip is cooled down to its stable operating temperature. It requires at least two measurements of a homogeneous target (e.g., non-reflecting plate or black body) providing different intensities to determine the gain of each spatial pixel. Afterwards, by fitting the pixel-dependent absolute radiometric calibration function to this pixel-wise gain, the response functions of all spatial pixels are homogenized among each other, which removes noise and stripes from the image.

The different intensities provided by the target can be realized either by a black body set to two different temperatures or by using two different integration times, which artificially change the intensity recorded by the sensor. During EUREC[4]A, the latter option was chosen. One set of images per channel was recorded with a low integration time of 10 µs (low radiance, simulating a cold target) and a second one with a higher integration time of 200 µs (high radiance, simulating a hot target). The

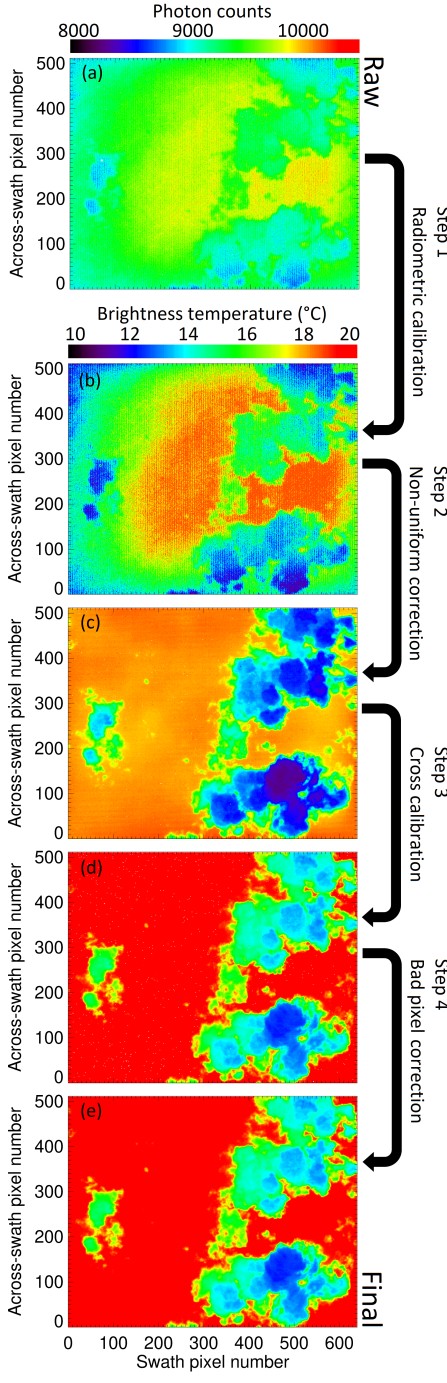

**Figure 2.** Illustration of the post processing steps using a 2D VELOX image acquired with the broadband channel (7.7 –12 μm) during the EUREC[4]A flight on 13 February 2020 at 11:37:30 UTC.

measurements were then performed with an integration time in between these maximum and minimum integration times (70 µs during EUREC[4]A).

Figure 2c shows the resulting image after applying the internal radiometric calibration model and the non-uniform correction, which significantly reduced the noise of the image.

### 3.2.3 Radiometric cross-calibration for field operation

The internal absolute radiometric calibration of the TIR imager is fixed unless no new radiometric calibration is performed in the labs of the manufacturer. It cannot be adapted during field operations and, therefore, it cannot account for changes of the system performance in a different environment. Only absolute offsets in the measured signal can be corrected for. To monitor the stability of the radiometric calibration with respect to the instrument aging and to adapt it to the environmental conditions (installation in the pressurized tube) during the research flights, cross-calibrations were performed with a mobile black body unit. In the laboratory, the black body was set to a temperature range comparable to the observations during EUREC[4]A. The environmental conditions (comparable filter and instrument internal temperature) in the laboratory were controlled to be similar to the flight conditions.

By operating VELOX without the window of the mounting tube, cross calibration factors are calculated by comparing pairs of the observed brightness temperature and the true brightness temperature generated by the black body. Tests at different environmental and target temperatures have shown that this adjustment is in a range of less than 1 K. Knowing the conditions during the flights and estimating the related offsets in advance, this step can be also directly combined with the internal radiometric calibration model introduced in Sect. 3.2.1.

Figure 2d shows the effect of the cross calibration. Compared to Fig. 2c, the brightness temperature is higher, which indicates the removal of a negative bias by the cross calibration.

### 3.2.4 Bad-pixel replacement

Most of the noise from the raw images is removed by applying the non-uniform correction. However, a second source for noise in the images is bad pixels (Budzier and Gerlach, 2015), which are caused by the manufacturing process of the sensor and the degeneration of the detector with time. Therefore, corrupt pixels are determined in the laboratory by directing the imager on a uniformly tempered black body. Pixels that show values exceeding the two times sigma variability of the entire image are classified as bad pixels. During the image processing, the bad pixels are replaced by the weighted average value determined by their four neighboring pixels.

Figure 2e shows the image after all bad pixels have been replaced. Although the total amount of detected bad pixels is low with less than 0.5 % (1,540 pixels) of the total pixel number (327,680), the correction significantly reduces the remaining noise.

### 3.2.5 Correction of window influence during airborne operation

For the airborne operation of VELOX in the tubular mounting, the effects of the window additionally need to be corrected. In this case, the observations are affected by transmission, absorption/emission, and reflection off/on the window surface. Transmission losses reduce the raw signal, while emission from the window and reflection on the inside of the window will increase it. Typically, the window is cooled down by the outer environment, which leads to a difference between the emission by the window (low temperature) and the reflected emission by the imager lens (higher temperature).

The radiance $I_{\mathrm{meas}}^{\uparrow}$ at flight altitude, which impinges on the outside of the window, will be mostly transmitted and the remaining fraction will be reflected or absorbed. Re-emission of absorbed radiation by the window occurs due to its own temperature $T_{\mathrm{win}}$. The resulting radiance $I_{\mathrm{meas}}^{\uparrow}$ behind the window is then composed of the transmitted radiation and the emitted radiation by one side of the window but also by reflected radiation, originating from the imager's lens emitted with its own temperature $T_{\mathrm{lens}}$. According to this, the following equation is applied for the correction of the measured signal:

$$I_{\mathrm{cor}}^{\uparrow} = \frac{I_{\mathrm{meas}}^{\uparrow} - \epsilon_{\mathrm{win}} \cdot I_{\mathrm{win}}(T_{\mathrm{win}}) - \epsilon_{\mathrm{lens}} \cdot I_{\mathrm{lens}}(T_{\mathrm{lens}}) \cdot \mathcal{R}_{\mathrm{win}}}{\mathcal{T}_{\mathrm{win}}}, \tag{1}$$

where $I_{\mathrm{cor}}^{\uparrow}$ is the corrected signal, using $I_{\mathrm{meas}}^{\uparrow}$ as input for the measured signal. The specific absorption/emission is given by $\epsilon$, the reflection by $\mathcal{R}$, and the transmission by $\mathcal{T}$. The subscripts denote the two coefficients of the window (win) and the imager's lens (lens). The spectral $\epsilon$, $\mathcal{R}$, and $\mathcal{T}$ of the window as provided by the manufacturer and validated by cross-calibrations with a black body are shown in Fig. 3. Overall, the Germanium window has a high average transmissivity of 93.95 % in the wavelength range from 7.7 µm to 12 µm. The spectral behavior of the reflection coefficient is rather constant over the entire range with about 5 % on average, while the absorption/emission coefficient is almost negligible for the VELOX channels 2 and 3, but affects longer wavelengths (up to 10 % for Channel 5 and 6). The emission coefficient of the lens is 0.15. Although, this value seems to be quite large, it results in a rather low contribution to the composed signal ($\approx 0.75$ %), because it only corresponds to the radiation emitted by the lens. For the application of Eq. 1 the window parameters were integrated for the filter response function of the selected spectral channel.

The window temperature is observed by a PT-100 temperature sensor attached to a part of the window that is not covered by the imagers FOV. It is displayed as a color-coded time series for the single flights in Fig. 4a. The temperature of the lens is measured internally by the imager and is displayed in Fig. 4b. Both show a dependence on flight altitude and environmental temperature, which has a significant time lag. For the first two hours of each flight, the airflow around HALO continues cooling down VELOX. However, once HALO has reached a stable flight altitude, the temperature changes remain small. The measurements thus benefit from maintaining a constant flight altitude.

### 3.3 Sensitivity and stability

A typical measure for the accuracy of measurements in the TIR wavelength range is the Noise Equivalent Temperature Difference (NETD). For the stand alone TIR imager, the manufacturer reports the NETD to be 40 mK. For the full setup, including

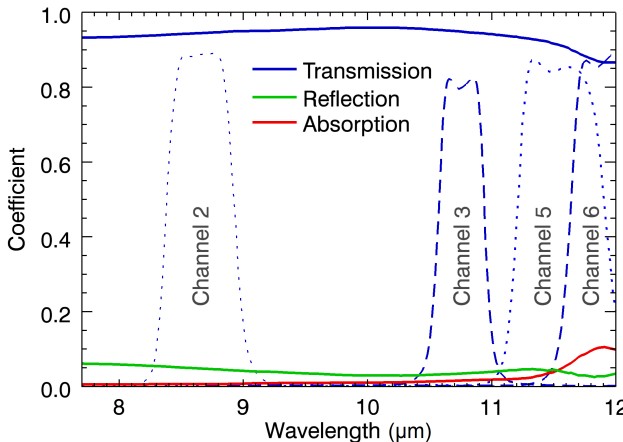

**Figure 3.** Spectral transmission, reflection, and absorption/emission coefficients of the Germanium window for the wavelength range covered by the TIR imager. Included are in addition the response functions (transmission coefficients, dashed/dotted lines) of the four narrow-band channels.

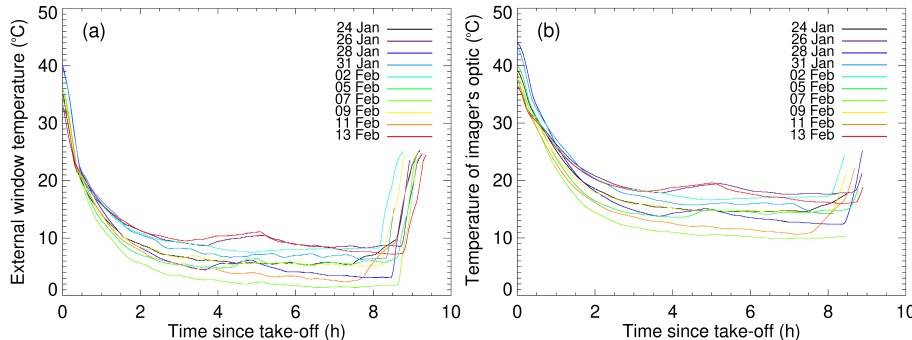

**Figure 4.** Time series of **(a)** the window temperature and **(b)** the temperature of the imager optics acquired during different EUREC$^4$A research flights.

spectral filters and the window, the NETD was estimated for each channel by measurements in front of a black body. The measurements are started after a two-hour spin up time to guarantee a stabilized filter and instrument operating temperature. Afterwards, 64 subsequent images at 10°C, 128 subsequent images at 20°C, and 64 subsequent images at 30°C were recorded. The difference between the average counts per pixel measured at 30°C and 10°C divided by 20 gives a counts-response file per degree Kelvin. From the 128 images measured at 20°C, the standard deviation of each pixel is calculated leading to a temporal noise file. Dividing the temporal noise file by the response file, averaging over all pixels, and multiplying it by 1000 gives the NETD in mK. The acquired NETDs are 48 mK for the two broadband channels (Channels 1 and 4), 347 mK for Channel 2, 605 mK for Channel 3, 473 mK for Channel 5, and 442 mK for Channel 6. Due to the use of the spectral filters, the detector

receives less radiation but is limited by the same thermal noise leading to the partly large NETDs. However, as long as the image values provide a significant contrast, the acquired thermal signal allows to resolve spatial structures of the target.

To quantify the absolute accuracy, 5 min image sequences were recorded, each using a different black-body temperature setting (5 K steps, -10 °C to 35 °C). Figure 5 shows box and whisker plots calculated from the mean measured brightness temperature of each single image for all six channels. On average, the measured brightness temperature matches well with the
320 black body generated temperature values. Furthermore, a comparison of the single sections reveals that the standard deviation is rather independent of the target temperature (no temperature-dependent trend in the boxes and whiskers).

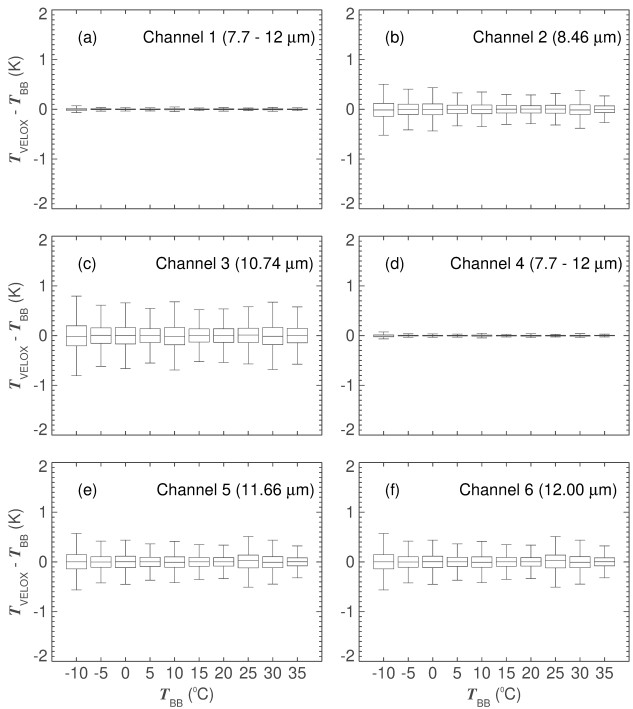

**Figure 5.** Box and whisker plots for each of the six channels of the TIR imager. Each panel, **(a)** to **(f)**, shows the deviations of the measured brightness temperature ($T_{\mathrm{VELOX}}$) of the single channels from the different black body temperature settings ($T_{\mathrm{BB}}$). The box encloses the interquartile range, defined at 25th and 75th percentile. The whiskers extend out to 1.5 times either the 25th or 75th percentile.

Averaged over the whole time series including all ten black body temperature settings, a standard deviation of 0.016 K is derived for both broadband channels. For the four other channels, the standard deviations are larger with 0.14 K for Channel 2, 0.23 K for Channel 3, 0.17 K for Channel 5, and 0.16 K for Channel 6. However, considering the 25th and 75th percentiles only,
the deviations are in the range of $\pm\,0.01$ K for the two broadband channels, $\pm\,0.09$ K for Channel 2, $\pm\,0.15$ K for Channel 3, $\pm\,0.11$ K for Channel 5, and $\pm\,0.1$ K for Channel 6.

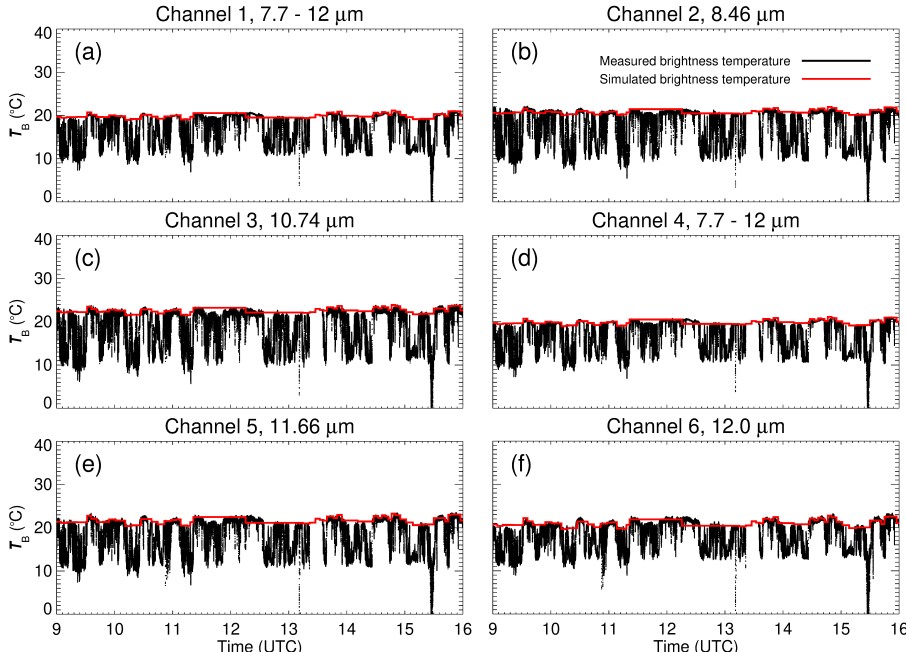

**Figure 6.** Time series of brightness temperature observed by the central 10 by 10 pixels of the thermal imager during the EUREC[4]A flight on 13 February 2020. Each panel, **(a)** to **(f)**, displays the brightness temperature observed by the use of the different channels (black) and the corresponding cloud-free simulations (red).

## 4 Measurement examples

During EUREC[4]A, VELOX was operated during 13 research flights leading to a total recording time of 104 hours, which results in approximately 82,000 km of horizontal cloud and surface observations. These raw data have been filtered for data quality, resulting in 85 hours (67,000 km) of suitable recordings.

Brightness temperatures measured by VELOX provide the potential for a large number of applications in remote sensing of the atmosphere and surface. A first set of basic retrieval products is presented in the following sections, including the cloud fraction and cloud top altitude.

### 4.1 Brightness temperature

Figure 6 displays time series of the averaged brightness temperature of the central 10 by 10 pixels (black) measured with the different channels of VELOX along the flight track on 13 February 2020. The highest values of the particular brightness temperature along the time series are related to cloud-free areas below HALO, indicating the detection of the brightness temperature of the ocean surface. All lower values are related to observations of the colder clouds.

Cloud-free simulations (red, see Appendix A) are used to validate the brightness temperature measurements for cloud-free areas. It can be nicely seen that the highest brightness temperature values along the flight track match the cloud-free simulations.

Figure 7 shows the correlation between the cloud-free simulations and the maximum brightness temperature values classified by a maximum envelope fit for all six channels. The maximum envelope fit is based on the time series of the averaged brightness temperature of the central 10 by 10 spatial pixels. The time series is divided into sections of 60 seconds, which was found to be the best setting for trade wind clouds. For each section, the maximum brightness temperature is used to build the envelope by

345 setting all measurements to this maximum value. If a 60-second sequence is fully covered by clouds, the maximum values of the previous cloud-free sequence is used for the envelope. This is justified, because temperature changes of the ocean surface can be assumed to be spatially (temporally) weaker compared to the effect of clouds. A 60-second sequence is defined fully cloudy, if its maximum brightness temperature is reduced by more than $3\%$ compared to the previous sequence. It should be noted that while this approach works well only for the trade wind cumuli observed during EUREC[4]A, it may fail for scenes

covered by extensive stratiform clouds.

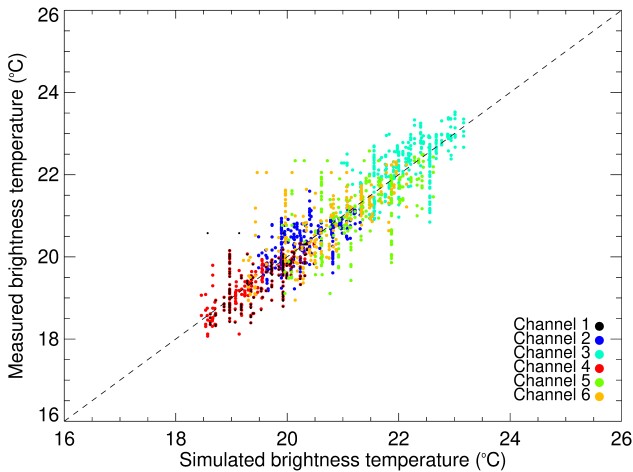

**Figure 7.** Correlation between the cloud-free simulations and the maximum brightness temperature values displayed in Fig. 6, which represent cloud-free measurements over the warm ocean.

The correlation $R$ between the maximum envelope fit and the simulations is 0.84 (Pearson correlation). Tests have shown that the measurement uncertainties have a minor influence on these differences. Although the NETD varies significantly between the single channels, the spread of the deviations observed with the different channels shows a similar pattern. The reason for the observed differences is mainly linked to the spatial/temporal resolution of the simulations, which are limited by the frequency

of dropsonde releases (between one sonde per 5 min and per 1 h). Within such a period, the simulations remain constant, while in reality, the atmosphere might change continuously. Excluding flight sections with low dropsonde density (e.g., 11:30 UTC to 13:30 UTC in Fig. 6) removes the largest differences, which supports this assumption. However, none of the six channels show a significant deviation from the general correlation. This indicates that the wavelength-dependent window transmissivity is well captured by the correction.

Figure 8 shows a series of 2D images of brightness temperature acquired by the broadband channel (Channel 1, 7.7 –12 $\mu$m) of VELOX during the EUREC[4]A flight on 5 February 2020. With HALO flying at 10 km altitude, the images cover an area

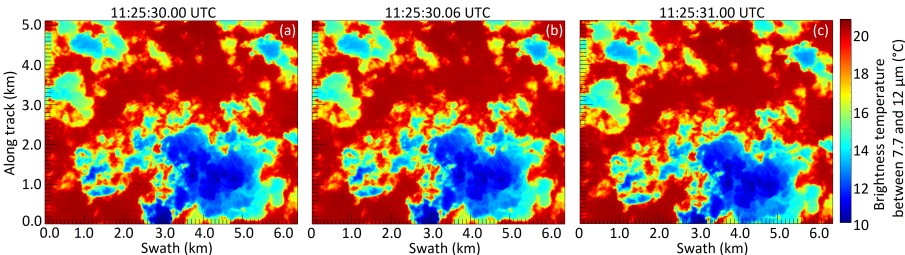

**Figure 8.** Two-dimensional fields of brightness temperature measured at a flight altitude of approximately 10 km with the VELOX broadband channel between 7.7 µm and 12 µm during the EUREC[4]A field campaign on 5 February 2020 at **(a)** 11:25:30.00 UTC, **(b)** 11:25:30.06 UTC, and **(c)** 11:25:31.00 UTC.

of 6.4 km by 5.1 km with a spatial resolution of 10 m by 10 m, both with respect to the surface. The images show the contrast between the warm ocean in red colors and the colder clouds in blue colors and highlight that VELOX is capable of resolving cloud structures with sizes well below 100 m. This is especially beneficial for observations in the trade wind region, where Schnitt et al. (2017) reported that 70 % of all observed clouds have a horizontal extension smaller than 2 km, which is further quantified for the present measurements from EUREC[4]A by Mieslinger et al. (2021).

VELOX is measuring with a frequency of 100 Hz. Therefore, each sequence of six channels is repeated after 0.06 sec, which translates into a flight distance of approximately 13 m for a typical flight speed of $220 \, \text{m s}^{-1}$. This means that subsequent images acquired with the same filter overlap as illustrated in Fig 8a and b. This overlap depends on the flight speed and the distance to the target. For the example shown in Fig. 8a and b, almost no shift of the scene is visible because 99.5 % of the scene is covered in both images. Therefore, Fig. 8c shows the acquired scene 1 sec later. Here, the shift becomes visible, but 91 % of the image overlaps with Fig. 8a.

This overlap opens additional options for analysing the VELOX images. Subsequent images with sufficient overlap provide a sequence of observations at different angular directions. Such image sequences have potential for stereoscopic 3D reconstruction of clouds (Kölling et al., 2019) and the investigation of 3D radiative effects in the TIR wavelength range. It further allows to study the small-scale geometry of shallow cumulus and dynamic processes at cloud edges, where the transition from water vapor to cloud droplets and entrainment take place.

To combine different spectral channels, the successive images need to match pixel wise. This requires to account for the horizontal shift between the images. This shift is estimated from the aircraft speed and the projected pixel size, while the latter depends on the distance between the surface/cloud and the aircraft (see Sect. 3.1).

For combining with traditional pushbroom imagers (specMACS; Ewald et al., 2015) or nadir pointing active remote sensing (WALES, HAMP; Wirth et al., 2009; Mech et al., 2014; Konow et al., 2019), the VELOX data was converted into a pushbroom-like data set by extracting the central swath of each image along the flight path.

Figure 9 shows a pushbroom image of brightness temperature measured at channel 1 for a 5 min time period on 5 February 2020. The scene covers the single images, which are shown in Fig. 8 and marked here by a black rectangle. Comparing the

real 2D images with the pushbroom-like images may indicate if fast moving clouds are distorted in the pushbroom view, which might have consequences for determining the cloud fraction from push-broom images (Konow et al., 2021).

Further applications for such a combination between VELOX and, e.g., specMACS are cloud retrievals that are based on measurements in the solar and thermal wavelength range and the investigation of the differential surface warming due to simultaneous cloud shadowing and cloud base emission.

## 4.2 Cloud cover

If the contrast between cloud top temperature and surface temperature is sufficiently high, TIR images are suitable for determining a cloud mask and to estimate the cloud fraction of a scene. In the case of measurements above open ocean, the cloud mask can be determined by a threshold method. During EUREC$^4$A, the ocean had a rather uniform horizontal temperature distribution at about $27 \pm 1°$C and provides a good background to identify even small and thin clouds. Even a slightly colder brightness temperature can identify a cloud. However, due to atmospheric absorption and emission, the brightness temperature of the ocean surface measured at the flight altitude of about $10\,$km is lower than the real ocean skin temperature. As the atmospheric conditions along the flight track can change, this needs to be considered when deriving the cloud mask by simple threshold methods. For example, for the flight on 9 February 2020, the measured brightness temperature of the cloud-free ocean varied between $19°$C and $21°$C (at $10\,$km flight altitude). Figure 10a shows the corresponding time series. To account for such fluctuations of the background brightness temperature, the envelope method described in Sect. 4.1 is applied to continuously adapt the cloud mask algorithm to changes in the atmosphere and ocean conditions by the maximum measured brightness temperature in the surroundings of the particular point of interest. As illustrated in Fig. 10a, the envelope reflects the circular flight pattern typically flown during EUREC$^4$A. This could be processed further assuming that spatial differences in sea-surface temperatures along a circle were/are larger than temporal variations over the course of a flight. For the cloud-mask algorithm, the difference between the measured brightness temperature and the envelope is calculated.

The reference level for measurements over cloud-free ocean is constant for the entire flight (Fig. 10b). Thus, a single threshold decides if a cloud was present or not. To account for the fuzzy nature of cloud edges, four different thresholds of $0.5\,$K, $1.0\,$K, $1.5\,$K, and $2.0\,$K were applied. For the flight shown in Fig. 10, this approach results in a total cloud fraction of $29.4\,\%$, $18.3\,\%$, $15.3\,\%$, and $13.5\,\%$ depending on the selected threshold.

Furthermore, the four thresholds were combined into a cloud mask. If the largest threshold of $2\,$K is exceeded, the measurement is denoted as "most likely cloudy", and if only the lowest threshold of $0.5\,$K is exceeded, the measurement is denoted as "probably cloudy". The state "cloud free" is classified when no threshold is reached, and "unknown" is classified when the cloud mask algorithm fails, e.g., due to condensation on the window. For the flight shown in Fig. 10, the combined cloud mask identifies $13.6\,\%$ of the data as "most likely cloudy", $17.9\,\%$ as "probably cloudy", and $68.5\,\%$ as cloud free.

The maximum envelope as derived from the central 10 by 10 spatial pixels, served as a reference in the cloud mask algorithm described above. Assuming that the envelope value is valid for the full 2D VELOX images, a 2D cloud mask and cloud fraction for each image was derived. Figure 11a shows the brightness temperature of a 2D VELOX image acquired during the EUREC$^4$A flight on 9 February 2020, and Fig. 11b shows the derived, combined cloud mask. It is obvious that the cloud mask

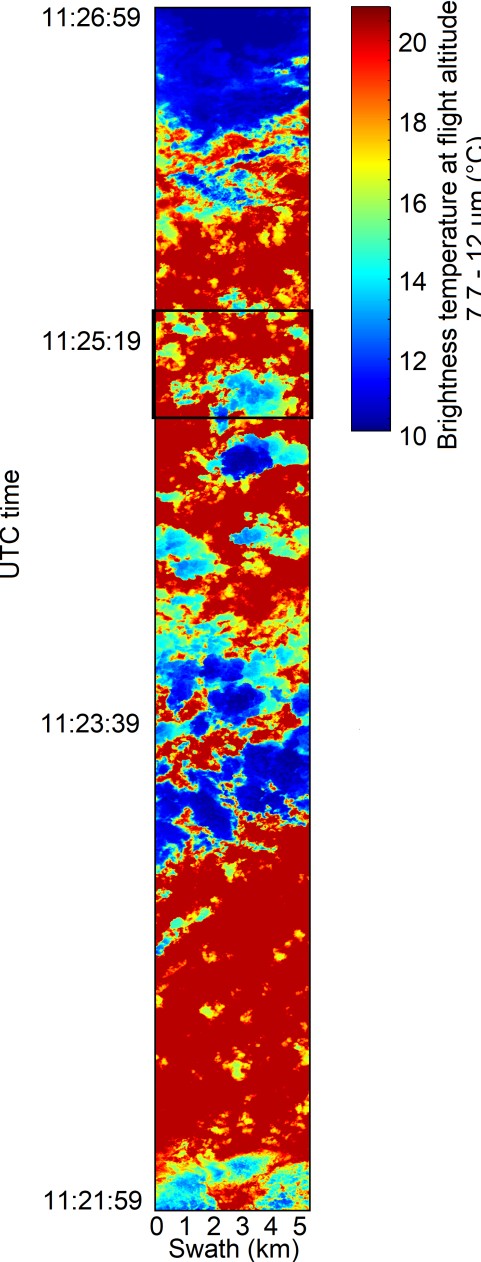

**Figure 9.** Two-dimensional time series of measured brightness temperature at a flight altitude of approximately 10 km using the center swath line of the VELOX broadband channel (7.7 µm to 12 µm). The data were acquired during the EUREC[4]A field campaign on 5 February 2020 between 11:22:00 UTC and 11:27:00 UTC. The black rectangle marks the measured area displayed in Fig. 8. Adapted from Konow et al. (2021).

.

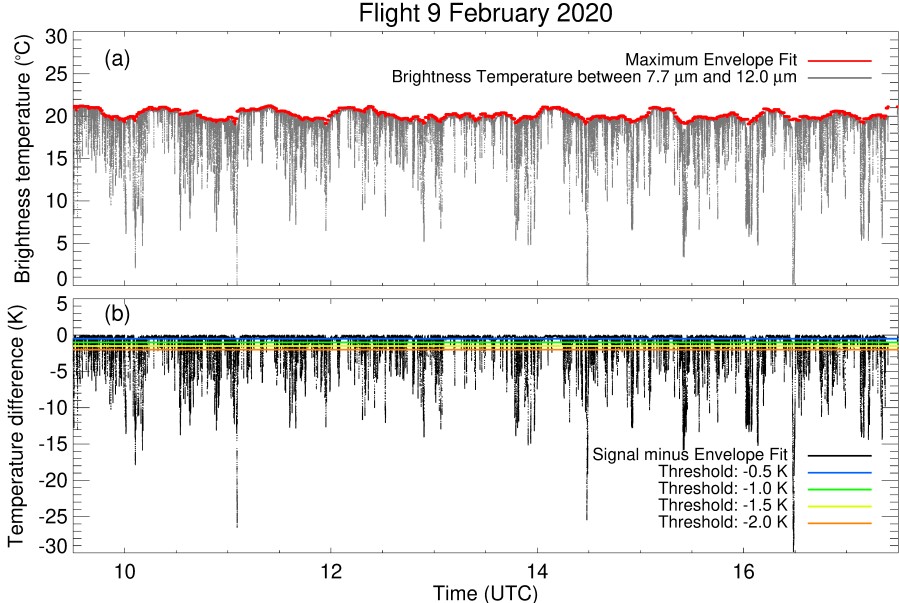

**Figure 10. (a)** Time series of brightness temperature (black) observed by the central 10 by 10 pixels of VELOX using Channel 1 during the EUREC[4]A flight on 9 February 2020. The maximum envelope fit is illustrated by the red dots. **(b)** Difference between the measured VELOX brightness temperature and the maximum envelope fit (black). The four thresholds of 0.5 K, 1 K, 1.5 K, and 2.0 K are included by color-coded lines.

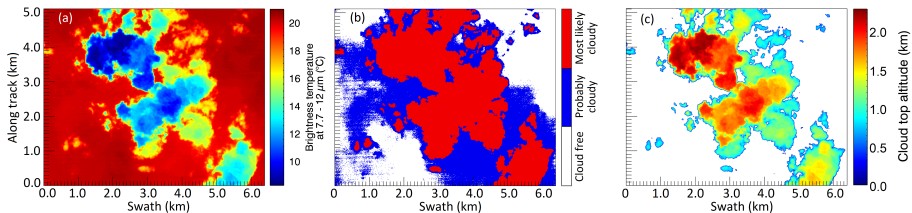

**Figure 11. (a)** Two-dimensional field of brightness temperature measured at a flight altitude of approximately 10 km with the VELOX broadband channel between 7.7 μm and 12 μm during the EUREC[4]A field campaign on 9 February 2020 at 15:05:21 UTC. For the same scene, panel **(b)** shows the combined cloud mask and panel **(c)** the retrieved cloud top altitude.

is also able to detect cloud patches smaller than 100 m in size. By integrating over all pixels in this scene, the cloud cover results in a total amount of 37.2 % "most likely cloudy". The amount of pixels identified as "probably cloudy" is 32.8 % and illustrates the uncertainty of the cloud mask algorithms, which always suffer from the fact that clouds are a dispersion, whose boundaries are often poorly defined..

     For EUREC[4]A, time series of the cloud masks and cloud fractions are available for all flights between 24 January and 15
February 2020 (Schäfer et al., 2021a, b), which were partly analysed and compared to other active and passive remote sensing

products by Konow et al. (2021). The comparison, based on the "most likely cloudy" threshold, highlighted the different sensitivities of the instruments to detect clouds, with VELOX always showing slightly larger cloud fractions compared to the other instruments (Konow et al., 2021). This illustrates the capability of VELOX to also detect small cloud patches. This is a benefit compared to most satellite products, which often suffer from detection limits that miss tiny low clouds (Someya and Imasu, 2018).

## 4.3 Cloud top altitude

The cloud top temperature measured by VELOX is closely linked to the cloud top altitude. This relation is commonly used in cloud top altitude retrievals from satellite observations. Here, a similar approach is used for the images from VELOX and extended by a cross-calibration with nadir-pointing cloud top altitude measurements from WALES (Wirth , 2021). This method allows to extend the nadir measurements of WALES to 2D maps of cloud top altitudes, which resolve the horizontally structure of shallow cumulus. To apply the cross-calibration, a first guess of cloud top altitude from VELOX is needed. It is derived from the measured brightness temperature of the thermal imager's broadband channel 1. This first guess is necessary since there is no fixed direct relation between cloud top altitude derived from WALES and the VELOX brightness temperature along the flight path. It rather varies in time with the changing influence of the atmosphere. For the first guess, the brightness temperature is combined with atmospheric profiles from dropsondes (George et al., 2021) and radiative transfer simulations of the cloud-free atmosphere. The simulated brightness temperatures are parametrized as a function of the distance to the cloud top and used to invert the measurements.

In a second step this first guess of the VELOX cloud top altitude is cross-calibrated with the WALES cloud top altitude. The cross-calibration uses the cloud mask ("most-likely-cloudy" threshold) of VELOX (cloud mask based on the central 10 by 10 spatial pixels) and WALES. If both instruments detect a cloud, the cross-calibration is applied, which links the first guess of the VELOX cloud top altitude to the WALES cloud top altitude in a linear relationship. At this juncture, the correction of the first guess VELOX cloud top altitude ranges between 100 m and 300 m.

Two major reasons for these uncertainties were identified; (i) an increased distance to the next dropsonde leads to uncertainties in the cloud-free simulations, and (ii) missmatches in the cloud mask by VELOX and WALES. The latter can be reduced when using the full temporal resolution of VELOX.

Considering the NETD of VELOX, the full approach allows a retrieval of 2D maps of cloud top altitudes with a vertical resolution of 40 m. As an example, Fig. 11c shows the derived cloud top altitude for the cloud scene from 9 February 2020. Cloud top altitudes below 600 m might be nonphysical and are related to very thin clouds or cloud edges. These low cloud top altitudes probably results from a contamination of the signal by the emission of the ocean below.

## 5 Conclusions

The technical setup, calibration, and correction procedures, and selected atmospheric applications of the new airborne, six-channel (7.7 $\mu$m to 12 $\mu$m wavelength) thermal infrared TIR imager VELOX (Video airbornE Longwave Observations within

siX channels) are presented. It is currently applied on board the High Altitude and LOng Range Research Aircraft (HALO), but may also be deployed on other airborne platforms or used for ground-based observations. VELOX provides spatially highly resolved 2D fields of upward terrestrial radiance or brightness temperature (FOV of 35.5° by 28.7°, 640 by 512 spatial pixels with 10 m by 10 m pixel size at a target distance of 10 km). The available spectral channels enable to retrieve 2D maps of cloud and surface properties such as cloud cover, cloud top altitude, phase, liquid/ice water path, effective radius, surface temperature, and snow/ice properties.

The post processing of VELOX measurements includes the application of an advanced radiometric calibration model including the effect of the window of the instruments housing during airborne observations, non-uniform corrections, cross-calibrations, and bad-pixel corrections. It was shown that significant additional efforts beyond the manufacturer calibration are needed to obtain reliable measurements of brightness temperature from VELOX during airborne operation. Recording the temperature in the instruments housing appears essential to track and consider the thermodynamic behavior of the entire system. The VELOX measurements were validated for cloud-free conditions over the ocean using radiative transfer simulations. The uncertainty analysis of the TIR imager revealed a high accuracy for the two broadband channels and slightly less accurate data for the other four narrow-band channels, where the detector receives less radiation but is limited by the same thermal noise. The Noise Equivalent Differential Temperature (NEDT) was estimated to be in the range of 48 mK for the broadband channels to 605 mK for Channel 3, and was found to be independent of the target temperature.

First measurements of VELOX, collected during the EUREC[4]A campaign were presented. Two-dimensional images of upward TIR radiance and brightness temperature, as well as time series along selected pixels and push-broom-like images were discussed. It was shown that all three methods provide detailed views on clouds and surfaces properties in a horizontal size range below 100 m. To validate the observations, comparisons with radiative transfer simulations were performed, which yield agreement within the range of measurement uncertainties. However, the agreement decreased with increasing distance to the closest dropsonde, and, thus, too large differences in the simulation input. Using maximum envelope fits and a threshold method, cloud fractions were calculated, which were applied to derive a combined cloud mask with confidence levels of "most likely cloudy", "probably cloudy", "cloud-free", and "unknown". For the identified cloudy parts, the cloud top temperature was resolved with a thermal resolution of better than 0.1 K, which translates into a resolution of approximately 40 m using basic cloud top altitude retrievals.

From the EUREC[4]A campaign, measurements of warm and liquid low-level trade wind cumuli over a warm ocean surface were available. It is envisioned to apply VELOX in three upcoming measurement campaigns; HALO-(AC)[3] operated out of Kiruna, Sweden in March/April 2022 (Wendisch et al., 2021), the EarthCARE Validation Campaign (ECVAL) in 2024, and the Tropical Ocean and Organized Convection (TOOC) campaign in 2024. These three measurement activities will provide data of different types of clouds (liquid, ice, mixed-phase stratus and convective clouds) in different climatic regions (Arctic, mid-latitudes, tropics). For these future campaigns it is planned to extend the existing products derived from VELOX (brightness temperature, cloud top temperature, cloud top altitude) to cloud-retrieval products (cloud phase, effective radius, liquid/ice water path) and surface properties (sea-surface temperature, snow/ice/water discrimination, snow grain size retrieval). For this purpose, the synergy with other instruments deployed on HALO will be very helpful.

*Data availability.* The brightness temperature fields from the TIR imager are published with a temporal resolution of 1 Hz (Schäfer et al., 2022) on the AERIS atmosphere Data and Services Centre (https://en.aeris-data.fr/; last access: 7 February 2022), which is part of the French Data Terra Research Infrastructure. Time series of the infrared thermometer observations are published on the AERIS EUREC[4]A Operational Center (https://observations.ipsl.fr/aeris/eurec4a/#/; last access: 7 February 2022). The cloud mask products from the TIR imager and from the infrared thermometer are published on AERIS by Schäfer et al. (2021a, b). All other data used and produced in this study are available upon request from the corresponding author (michael.schaefer@uni-leipzig.de).

## Appendix A: Cloud-free radiative transfer simulations

To validate and process the measurements obtained with VELOX during EUREC[4]A, radiative transfer simulations are performed along the flight path of HALO (position, altitude, and time) using libRadtran (Emde et al., 2016). Spectral upward radiance is determined with version two of the DIScrete ORdinaTe (DISORT 2.0) solver (Stamnes et al., 2000) using the 16-stream approximation and is directly converted into brightness temperature. Internally, a spectral resolution of 1 nm is applied and interpolated on the channel wavelength ranges of VELOX (see Tab. 2). Molecular absorption is considered by the representative wavelength approach (REPTRAN), described by Gasteiger et al. (2014). Ozone ($O_3$) absorption is implemented and accounts for the strong $O_3$ absorption band at 9.5 µm to 10.5 µm wavelength. Merged profiles of radio soundings from the Barbados Cloud Observatory (BCO, Stevens et al., 2016) with dropsonde measurements from HALO (George et al., 2021) represent the atmospheric conditions above and below the aircraft. For each simulation, the dropsonde closest in time to the location of HALO was selected. The sea-surface temperature $T_{sea}$, which was rather constant over the EUREC[4]A period, is set to 300 K (26.85°C), according to the observations by the research vessels, e.g., the Maria S Merian or the Ron Brown, the deployed saildrones, and buoys during the entire campaign. The ocean emissivity ($\epsilon_{ocean}$) is derived by subtracting a wavelength dependent ocean albedo ($\alpha_{ocean}$) file from unity ($\epsilon_{ocean} = 1 - \alpha_{ocean}$). The applied ocean-albedo file is provided by the International Geosphere Biosphere Programme (IGBP; Belward and Loveland, 1996).

*Author contributions.* MS and KW were the primary authors of the paper. BS and MW designed the airborne experiments, which were performed by MS, KW, AE, and MW. The technical work to construct the VELOX system was performed by CH, TR, and FJ. Simulations with libRadtran were mainly performed by KW, MS, and EJ. MS, KW, AE, and EJ analysed and compared the observations and simulations. JT performed the cloud mask analysis, JM the cloud top altitude analysis. AE, MW, EJ, and BS provided technical guidance. All authors contributed to the interpretation of results and wrote the paper.

*Competing interests.* The authors declare that they have no conflict of interest.

*Acknowledgements.* Very much, we thank the Max Planck Institute for Meteorology, Hamburg, Germany for the funding of the new VELOX system and providing it to the HALO community. We gratefully acknowledge the funding by the Deutsche Forschungsgemeinschaft (DFG, German Research Foundation) – Projektnummer 268020496 – TRR 172, within the Transregional Collaborative Research Center "ArctiC Amplification: Climate Relevant Atmospheric and SurfaCe Processes, and Feedback Mechanisms (AC)[3]. We are further grateful for funding of the project WE 1900/38-1 by the DFG within the framework of the Priority Programme SPP 1294 to promote research with HALO. Many

thanks to the German Aerospace Center (Deutsches Luft und Raumfahrtzentrum, DLR) for the highly appreciated support before, during, and after the EUREC[4]A campaign.

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
