# Peer review of "VELOX - A new thermal infrared imager for airborne remote sensing of cloud and surface properties"

_Atmospheric Measurement Techniques, 2021_

## Referee Comment (RC2)

Review of « VELOX - A new thermal infrared imager for airborne remote sensing of cloud and surface properties », by Michael Schäfer et al.

**General comments**

This paper presents a new thermal infrared (TIR) imager called VELOX aimed at measuring cloud and surface properties, primarily developed to be used onboard the High Altitude and Long Range Research Aircraft (HALO). VELOX is adapted from a commercial TIR imager and measures brightness temperature in 6 channels with 10 m spatial resolution at the surface when flying at 10 km. First the instrument hardware and technical characteristics are described, then the calibration procedure, which includes several successive steps, is detailed. The performances of the instrument in terms of noise-equivalent temperature difference (NETD) and absolute accuracy are assessed using a temperature-controlled blackbody reference. Finally, measurements acquired during the EUREC$^4$A campaign in 2020 are used to illustrate the capabilities of the instrument, with first hints to the derivation of a cloud mask and cloud top altitude from measured brightness temperatures.

The paper is well organized and well written, and perfectly fits with the scope of AMT. However, given that the focus is on the instrumentation and on the data processing, more details would be appreciated to clarify several technical points that so far remain somehow unclear. Only once these important clarifications have been made the paper could be considered for publication.

**Specific comments**

1) As very similar sensors have been (or are still) used in airborne configuration, it would be useful to highlight the specificities of VELOX. In particular, does the configuration respond to specific requests that no existing instrument would match? Do the performances enable improved retrievals?

2) As explained in the text, the successive images acquired by VELOX largely overlap. It is not clear whether this massively redundant information is useful or whether using larger integration times could advantageously improve the accuracy of the measurement. In any case the chosen acquisition configuration would deserve more justification.

3) In general the description of the calibration procedure is very qualitative, making hard for the reader to really guess what is practically done. Replication of the procedure would probably be quite difficult. More details (in particular for the correction of the window impact) would be helpful. Adding equations to more explicitly describe the successive steps would certainly help as well. Several suggestions are made in the technical corrections.

**Technical corrections**

l.10: analysis

l.22: is estimated wrong → estimation is wrong

l.23: not clear what *polar orbiting satellites* refers to. Is the sentence valid only for such orbits?

l.28: *comparable higher* sounds contradictory

l.31: information on overpasses should be merged with that at l.23

l.33: can you detail why MSI will be better than current sensors? More channels, higher spatial resolution?

l.35: *kilometer range* sounds similar to MODIS. Is it actually coarser?

l.51: *dominated by* is unclear. Radar are indeed not sensitive to small particles, but lidar are sensitive to large ones (although few large particles may reflect much less radiation than many smaller particles). Radar are mostly useful where lidar signal saturates

l.53: *different vertical weightings* is unclear. Are you talking about the profiling capability?

l.55: dominated → driven, governed?

l.63: "3D effects" also take place in the TIR. Maybe clarify what effects typical to the SW are avoided

l.65-68: this paragraph is not very clear. What is challenging? To measure a temperature difference, to know perfectly the reference (background signal)? Knowing the detector temperature is not enough, the whole instrument contributes to the measured signal. What is *not always given*?

l.69: when presenting a new instrument, it's useful to point how it differs from existing available (sometimes commercially) instruments, here or later on in the manuscript

l.73: repetition of reference for EarthCARE

l.93: what's the size of the filter wheel? Of the filters? As they do not appear in Fig. 1 I assume they're quite small

l.94: 100 Hz is for a complete rotation or to go from one filter position to the next? Clarify the link with the 100 Hz acquisition. A single frame on each filter or integration of multiple frames on the same filter? State here that the full measurement on all filters takes 0.06 s, this is a major information.

l.95: *partly adapted* is unclear. Are the filters meant to match MODIS filters characteristics or only the central wavelength?

Table 1: what is the temperature reference for the NETD? Could you provide more details in the text about the cooling of the sensor (temperature, stability etc.)

l.102: is the spectral response of the detector really zero outside of the range 7.7 – 12 microns?

l.102: maybe state here the reasons (if any) for duplicating this broadband channel

l.144: I'm not sure to get what the first issue is. Is it to project the pixels at the Earth surface? Is the aircraft movement used to tackle this, or just the position (including attitude)? Does it work when the emitter is not the surface, but a cloud?

l.169: can the difference in acquisition time for different filters be an issue as well? To be related with the distance traveled by the aircraft between two successive filters

l.175: this scene identification/matching deserves more details. What kind of algorithm is used?

l.189: In a system without on-board calibration, this calibration procedure is crucial. Can you provide more details on the way the corrections were obtained (lab experiments to isolate the impact of temperature changes?). At least consider referring to a paper detailing how this is achieved.

l.205: it is not clear what the link between *gain/offset* and *variable ground potentials* is. Once the non-uniformity of the pixels is identified, how is the amplitude of the correction determined? Why are stripes removed with this procedure? Do these stripes come from pixels with different gains or different offsets (due to straylight for instance)?

l.207: does the scene need to be homogeneous to apply this calibration, as stated above?

l.214: I don't understand why this specific calibration is not implemented directly at the step 3.2.1. Practically, is the correction pixel-dependent? Is it static or does it depend on environmental conditions?

Eq. 1: how where the different parameters of this equation determined? Was the method validated by cross-calibration against a black-body?

l.268: what is *accuracy* here? Absolute accuracy detailed just below?

l.297: how can you know that no cloud-free ocean was observed?

l.300: are the differences between simulated and estimated cloud-free BT due to differences in atmospheric state, or could they be solely explained by measurement uncertainty? The differences should be compared to measurement uncertainty on the one hand, and to simulation variability on the other hand. Are the points away from the 1:1 line actually those acquired far from a dropsonde?

l.317: what is the interest of such a comparison with pushbroom configuration, if the obtained differences are not better described?

l.347: for this, is the average of 10x10 pixels used or it is performed for individual pixels? Does the maximum envelope comes from the time series of individual pixels or from a single image?

l.368: here it is somehow assumed that the cloud is optically thick and that the emission comes from the top of the cloud. Can you discuss a bit these assumptions and their limits? Would changes in LWC or $r_{eff}$ make a difference on the emissivity fixed to 0.99?

l.376: 470 m offset seems huge for a cloud mostly ranging from 600 to 1400 m. Can it really be explained by errors in actual atmospheric profile? How does an error in BT translate into an error in cloud top altitude, roughly (for the atmospheric profiles observed)?

l.378: can you detail this correction procedure since it may be critical (when errors are larger than the measured range of variations)

l.396: references to EUREC[4]A not needed here

l.408: typo: *Oceanc*

l.433: how is set ocean emissivity?

---

## Author Comment (AC1)

**We thank the reviewer for her/his helpful comments, which certainly improved the manuscript. The detailed replies on the reviewer's comments are structured as follows: The individual reviewer comments are given in bold letters, followed by our reply. Changes/additions made to the text are enclosed in quotation marks.**

**l. 114: "The non-imaging infrared thermometer has a larger and more sensitive detector..." this is not obvious as Table 1 show a larger NETD value**

We thank the reviewer for pointing at this issue. We corrected it in a meaningful way.

*"The non-imaging nadir pointing infrared thermometer has a smaller FOV and a lower sensitivity than the TIR imager (compare Tab. 1). However, the sensor calibration is more stable, does not need to be cooled and, therefore, is well suited to serve as a reference for the TIR imager. The spectral window of the infrared thermometer covers the spectral channel 3 of the TIR imager (compare Tab. 2), what allows for a measurement comparison between the two instruments and cross-calibration checks."*

**l. 243, Eq. (1): What does the "imager optics" mean? The external lens of the optic is certainly transparent enough in the infrared to have a low emissivity and to let pass an important fraction of the radiation emitted by "the bottom" of the optic (the detector?). Are the imager and its optics (lens, detector, etc.) assumed to be isothermal? How are the temperature T_opt and the emissivity epsilon_opt determined?**

The imager optics refers to the lens of the imager, which also emits radiation. This emitted radiation is partly reflected back by the window and finally contributes to the detected signal. Although, the lens is designed to have a high transmissivity in the considered wavelength range, it is not perfectly transparent. Therefore, it absorbs and reemits radiation with its own temperature. Isotherm conditions in the camera, especially between lens and detector, are not given, because the detector is cooled to lower temperatures than the camera environment. The temperature of the lens is directly measured by the imager itself. Its emission coefficient is provided by the manufacturer. The different temperatures of the lens, the instruments body, and detector are all accounted for at some point in the calibration process (please see also our changes in Sect. 3.2.1). However, to avoid any confusion, we now call it "lens" and not "imager optics". We also renamed the parameter subscripts from "opt" to "lens". We made some further revisions to describe the sources of the different parameter values.

*"…, originating from the imagers lens emitted with its own temperature $T_{lens}$."*

*"… The specific absorption/emission is given by $\varepsilon$, the reflection by R, and the transmission by T. The subscripts denote the two coefficients of the window (win) and the imager's lens (lens). The spectral $\varepsilon$, R, and T of the window as provided by the manufacturer and validated by cross-calibrations with a black body are shown in Fig. 3. Overall, the Germanium window has a high average transmissivity of 93.95 % in the wavelength range from 7.7 µm to 12 µm. The spectral behavior of the reflection coefficient is rather constant over the entire range with about 5 % on average, while the absorption/emission coefficient is almost negligible for the VELOX channels 2 and 3, but affects longer wavelengths (up to 10 % for Channel 5 and 6). The emission coefficient of the lens is 0.15. Although, this value seems to be quite large, it results in a rather low contribution to the composed signal ($\approx$ 0.75 %), because it only corresponds to the radiation emitted by the lens. For the application of Eq. 1 the window parameters were integrated for the filter response function of the selected spectral channel.…"*

[Figure]

Figure 3: Spectral transmission, reflection, and absorption/emission coefficients of the Germanium window for the wavelength range covered by the TIR imager. Included are in addition the response functions (transmission coefficients, dashed/dotted lines) of the four narrow-band channels.

**l. 297: how is the "no cloud-free" condition determined?**

The highest brightness temperatures observed along the flight track are attributed to the cloud-free regions as we can assume that the highest temperatures are related to the warm ocean surface. Changes of the brightness temperature of the ocean surface are expected to be small in comparison to sudden temperature drops induced by clouds. Therefore, if the highest brightness temperature observed within a 60-second sequence is reduced by more than 3 % compared to the previous 60-second sequence, it is highly likely that clouds were present within this time frame. In this case, the calculated maximum brightness temperature envelope is set to the value of the previous cloud-free sequence. Using the 2D images, this method was visually validated for different cloud situations. We revised the part and added more information:

*"… If a 60-second sequence is fully covered by clouds, the maximum values of the previous cloud-free sequence is used for the envelope. This is justified, because temperature changes of the ocean surface can be assumed to be spatially (temporally) weaker compared to the effect of clouds. A 60-second sequence is defined fully cloudy, if its maximum brightness temperature is reduced by more than 3 % compared to the previous sequence. …"*

**l. 357: which of the two cloud fraction, "most likely cloudy" or "probably cloudy" is used for this comparison?**

The "most likely cloudy" cloud fraction is used for this comparison. We've now added this information in the text.

*"The comparison, based on the "most likely cloudy" threshold, highlighted the different sensitivities of the instruments to detect clouds, with VELOX always showing slightly larger cloud fractions compared to the other instruments (Konow et al., 2021)."*

**l. 376: I understand that the distance to the nearest dropsonde introduces errors, but why would these errors be systematic enough to generate a bias? I would rather expect a random error.**

The reviewer is right. Our explanations of the dropsonde uncertainties were a bit too sparse. Together with other comments related to the cloud top altitude retrieval, we revised the entire section. We now use the VELOX cloud top altitude only as a first guess, which is then directly cross-calibrated by the data from WALES. To answer your question in more detail:

These offsets are randomly distributed (see. Fig R1a). The mean of the offset distribution is located close to 0 m for all flights. The 25 % percentile is about 120 m. For a flight leg with low dropsonde coverage (see Fig. R1c), 50 % of the VELOX cloud top altitudes had an offset smaller than 400 m (for low temporal resolution of 1 Hz) in pointwise comparisons with WALES. This offset could be further reduced by the use of a temporally higher resolved time series (full VELOX resolution, Fig. R1b).

This improvement indicates that the offsets are affected by the differences between the VELOX and WALES cloud masks, especially due to mismatches at cloud edges. A second source for larger offsets is attributed to the distance to the closest dropsonde, which introduces uncertainties in the radiative transfer simulations and the parametrization of the cloud-free atmospheric temperature offset. This is confirmed by the analysis of flight sections with a low number of dropsonde launches (2 February 2020). In these cases, the offsets increase as illustrated in Fig. R1c.

[Figure]

Figur R1: For EUREC[4]A flight on 2 February 2020, **(a)** offsets beween cloud top altitude from VELOX and WALES, **(b)** Absolute offsets beween cloud top altitude from VELOX and WALES compared for 1 Hz VELOX data (red) and full VELOX temporal resolution (purple), and **(c)** offsets for flight sections with low dropsonde density. Red/purple lines indicate median values.

The revised section reads now like this:

*"The cloud top temperature measured by VELOX is closely linked to the cloud top altitude. This relation is commonly used in cloud top altitude retrievals from satellite observations. Here, a similar approach is used for the images from VELOX and extended by a cross-calibration with nadir-pointing cloud top altitude measurements from WALES (Wirth et al., 2021). This method allows to extend the nadir measurements of WALES to 2D maps of cloud top altitudes, which resolve the horizontally structure of shallow cumulus. To apply the cross-calibration, a first guess of cloud top altitude from VELOX is needed. It is derived from the measured brightness temperature of the thermal imager's broadband channel 1. This first guess is necessary since there is no fixed direct relation between cloud top altitude derived from WALES and the VELOX brightness temperature along the flight path. It rather varies in time with the changing influence of the atmosphere. For the first guess, the brightness temperature is combined with atmospheric profiles from dropsondes (George et al., 2021) and radiative transfer simulations of the cloud-free atmosphere. The simulated brightness temperatures are parametrized as a function of the distance to the cloud top and used to invert the measurements.*
*In a second step this first guess of the VELOX cloud top altitude is cross-calibrated with the WALES cloud top altitude. The cross-calibration uses the cloud mask ("most-likely-cloudy" threshold) of VELOX (cloud mask based on the central 10 by 10 spatial pixels) and WALES. If both instruments detect a cloud, the cross-calibration is applied, which links the first guess of the VELOX cloud top altitude to the WALES cloud top altitude in a linear relationship. At this juncture, the correction of the first guess VELOX cloud top altitude ranges between 100 m and 300 m.*

*Two major reasons for these uncertainties were identified; (i) an increased distance to the next dropsonde leads to uncertainties in the cloud-free simulations, and (ii) missmatches in the cloud mask by VELOX and WALES. The latter can be reduced when using the full temporal resolution of VELOX. Considering the NETD of VELOX, the full approach allows a retrieval of 2D maps of cloud top altitudes with a vertical resolution of 40 m. As an example, Fig. 11c shows the derived cloud top altitude for the cloud scene from 9 February 2020. Cloud top altitudes below 600 m might be nonphysical and are related to very thin clouds or cloud edges. These low cloud top altitudes probably results from a contamination of the signal by the emission of the ocean below."*

[Figure]

Figure 11. (a) Two-dimensional field of brightness temperature measured at a flight altitude of approximately 10 km with the VELOX broadband channel between 7.7 μm and 12 μm during the EUREC4A field campaign on 9 February 2020 at 15:05:21 UTC. For the same scene, panel (b) shows the combined cloud mask and panel (c) the retrieved cloud top altitude.

**l.404: the bias in the retrieved altitude should be mentioned here**

The reviewer is right. With the former method this would have been necessary information at this point. However, since we've revised the cloud top altitude section and now calibrate the VELOX data directly with WALES the offsets are not really treated as a bias anymore, they are rather a calibration factor. Therefore, we give the accuracy of 40 m here instead, which is related to the NETD of VELOX.

---

## Author Comment (AC2)

**Dear Quentin Libois, we thank you for the many helpful comments, which certainly improved the manuscript. Especially pointing out the small but important details, which were missing so far, was very welcome. The detailed replies on the reviewer's comments are structured as follows: The individual reviewer comments are given in bold letters, followed by our reply. Changes/additions made to the text are enclosed in quotation marks.**

**Specific comments**

1) **As very similar sensors have been (or are still) used in airborne configuration, it would be useful to highlight the specificities of VELOX. In particular, does the configuration respond to specific requests that no existing instrument would match? Do the performances enable improved retrievals?**

We agree, that the novel features of the instruments, which makes VELOX well-suited for atmospheric measurements need more discussion and revised the manuscript accordingly. Especially in conjunction with your later comment on line 69, we've included a comparison to existing instruments and extended the description of the major advantages of VELOX.

However, the primary objective of VELOX was not to have a new advanced instrument. It rather aims to complete the HALO remote sensing instruments suite to a complete EarthCARE like instruments setup (Stevens et al. 2019). Major requirements of VELOX have been to cover the thermal spectral channels of MSI on EarthCARE and from a practical point of view it had to fit into the limited space on HALO. Still, the advanced measurement technique provided some improvements that distinct VELOX from common thermal imagers. One specific feature of VELOX is that it does not need an on-board calibration. This is compensated by operating VELOX in a temperature stabilized housing and by a series of post-calibration routines as described in the manuscript. Compared to the common line scanners applied in airborne thermal infrared (TIR) observations, VELOX consists of a two-dimensional (2D) sensor and measures with a high frame rate of up to 100 Hz. This allows for a wide range of applications, like analysing the horizontal fine-scale structure of clouds and dynamic processes at clouds edges, stereoscopic image processing using the large overlap of the individual images, or quantifying differential warming of the sea surface in combination with solar imagers.

Choosing similar spectral channels like they are used on satellite IR instruments, VELOX does not aim to suite new retrieval techniques. For the analysis of the VELOX data it is currently planned to use existing retrieval techniques developed for EarthCARE or former satellite sensors like MODIS, AVHRR, or ASTER. However, together with the state-of-the-art HALO instruments suite it is expected to derive new synergetic retrievals together with lidar, radar, solar spectrometers, or microwave radiometers that were not possible on HALO before.

We tried to include this discussion in the revised manuscript. We pasted all off the related revisions made in the manuscript together with our answer to your comment referring to line 69.

2) **As explained in the text, the successive images acquired by VELOX largely overlap. It is not clear whether this massively redundant information is useful or whether using larger integration times could advantageously improve the accuracy of the measurement. In any case the chosen acquisition configuration would deserve more justification.**

The high temporal resolution and large overlap of successive images is intended for different reasons. VELOX obtains the six spectral channels using a filter wheel. To match images of different channels, when measuring on a fast flying aircraft, the measurement frequency needs to be fast. 100 Hz with 6 channels gives a frame rate of about 15 Hz. This high frame rates results in an overlap of the images, which we aim to use in the analysis. Measurements of the same target from different angular directions can be analysed with respect to 3D radiative effects or processed by stereoscopic methods

to reconstruct the cloud structure. To make this clearer, we've added the following paragraphs in Sects. 2.1 and 4.1.

*"To match images of different channels, when measuring on a fast flying aircraft, the measurement frequency needs to be fast. Therefore, the whole filter wheel rotates with a frequency of 100 Hz. With six channels, this results in a frame rate of about 15 Hz, meaning that a full filter-wheel spin to acquire separately six single images (one with each of the six filters) takes 0.06 sec."*

*"This overlap opens additional options for analysing the VELOX images. Subsequent images with sufficient overlap provide a sequence of observations at different angular directions. Such image sequences have potential for stereoscopic 3D reconstruction of clouds (Kölling et al., 2019) and the investigation of 3D radiative effects in the TIR wavelength range. It further allows to study the small-scale geometry of shallow cumulus and dynamic processes at cloud edges, where the transition from water vapor to cloud droplets and entrainment take place."*

Reducing the frame rate in order to increase the integration time wouldn't improve the quality of the images significantly. Due to the use of the spectral filters, which have an own temperature of about 20°C, we expect a higher contribution from the filters itself to the total measured signal. Using integration times of 50 μs to 70 μs, this effect can be confidently corrected for. For higher integration times, the filter signal becomes too large to be corrected for and contaminates the images. To clarify this, we've added the following text, when we introduce the integration time.

*"During EUREC$^4$A, the imager was operated with a frame rate of 100 Hz and an integration time of 70 μs, for which the thermal influence of the spectral filters can be still well corrected for."*

3) **In general, the description of the calibration procedure is very qualitative, making hard for the reader to really guess what is practically done. Replication of the procedure would probably be quite difficult. More details (in particular for the correction of the window impact) would be helpful. Adding equations to more explicitly describe the successive steps would certainly help as well. Several suggestions are made in the technical corrections.**

Thank you very much for pointing this out. Actually, many of the steps are automatically performed by the manufacturers data processing software, making it easy to replicate the measurements by any user. To let the reader know, what data processing is included in the manufacturer software, we also try to explain the procedures behind all the other single steps (e.g., the non-uniform correction) without aiming for a replication of the existing tools. Only some processing steps like the cross-calibration or window correction need more treatment.

To avoid any confusion, we revised the manuscript accordingly to make clearer, what is done by the software and what needs to be done afterwards. In this respect, your comments in the technical correction part were very valuable. You will find all of the revisions we've made together with our answers to the single questions below.

**Technical corrections**

**l.10: analysis**

Corrected to *"analysis"*

**l.22: is estimated wrong → estimation is wrong**

Changed to *"estimation is wrong"*

**l.23: not clear what *polar orbiting satellites* refers to. Is the sentence valid only for such orbits?**

You are right. At this point, the reference to polar orbiting satellites was confusing, but also the whole content of the sentence related to the temporal resolution was not fitting here as well and is repeated later again. Therefore, we've deleted the whole sentence. According to your fourth comment (l.31) we've included this information later.

**l.28: *comparable higher* sounds contradictory**

We've deleted the word "comparable".

**l.31: information on overpasses should be merged with that at l.23**

Changed. The new paragraph reads like this:

*"However, because all these sensors are operated on polar orbiting satellites their temporal sampling is limited."*

**l.33: can you detail why MSI will be better than current sensors? More channels, higher spatial resolution?**

Our former wording, when we wrote that MSI will provide a "much improved" replacement was misleading. For the EarthCARE satellite with all its instruments this might be true, but you are right, for MSI alone this argument doesn't hold. Although the spatial resolution of MSI is better than that of MODIS, it has a smaller swath, due to the lower orbit of EarthCARE. Furthermore, it has fewer channels compared to MODIS, although they are sufficient for the purpose the instrument was designed for. Therefore, we revised this sentence with the following:

*"The Multi-Spectral Imager (MSI; Illingworth et al., 2015) instrument on the proposed Earth Clouds, Aerosol and Radiation Explorer satellite mission (EarthCARE; Illingworth et al., 2015) will provide a valuable replacement for, e.g., MODIS. Although it provides fewer channels and covers a smaller swath, the spatial resolution will be increased. However, as a polar orbiting satellite, it will suffer from similar temporal sampling issues."*

**l.35: *kilometer range* sounds similar to MODIS. Is it actually coarser?**

You are right, kilometre range sounds a bit too bad. The TIR channels have a best spatial resolution of 2 km. We removed "kilometre range" and added the number.

*"…, the spatial resolution of the TIR channels is only in the range of 2 km (Schmit et al., 2018)."*

**l.51: *dominated by* is unclear. Radar are indeed not sensitive to small particles, but lidar are sensitive to large ones (although few large particles may reflect much less radiation than many smaller particles). Radar are mostly useful where lidar signal saturates.**

Yes, we've mixed up the facts by putting all of them in the same sentence. We've revised this paragraph as follows:

*"Based on the single-scattering assumption, active radar (e.g., the HALO Microwave Package, HAMP; Mech et al., 2014, Konow et al., 2019) observations are most sensitive to large cloud droplets and ice crystals and mostly can penetrate the entire cloud. Active lidar (e.g., the Water vapor Lidar Experiment in Space, WALES; Wirth et al., 2009) observations are also sensitive to the backscattering of small cloud droplets and therefore attenuated quickly in liquid clouds. In comparison, passive instruments …"*

**l.53: *different vertical weightings* is unclear. Are you talking about the profiling capability?**

Yes, this discussion aims to point at the profiling capabilities, but also intends to highlight that the different vertical weightings need to be considered for the interpretation of the different measurements, e.g., from where inside the cloud the signal originates. We revised it accordingly.

*"To interpret and compare solar and thermal observations, their different vertical weightings have to be considered, e.g., the thermal emission originates closer to the cloud top compared to solar radiation, which is to some degree also scattered from lower cloud layers (Platnick et al., 2000)."*

**l.55: dominated → driven, governed?**

Changed to *"governed"*

**l.63: "3D effects" also take place in the TIR. Maybe clarify what effects typical to the SW are avoided**

We've added this information in parantheses.

*"For example, three-dimensional (3D) radiative effects (shadowing, multiple-scattering, horizontal photon transport), …"*

**l.65-68: this paragraph is not very clear. What is challenging? To measure a temperature difference, to know perfectly the reference (background signal)? Knowing the detector temperature is not enough, the whole instrument contributes to the measured signal. What is *not always given*?**

We agree that the discussion was insufficient and revised this paragraph.

*"However, for TIR sensors the translation of the raw counts to physical units is challenging. Compared to imagers in the solar spectral range, radiation emitted by the target is not the only source of radiation, which affects the detector signal. The detector itself emits radiation based on its temperature and in general does register a temperature difference between the target and the detector temperature. Therefore, keeping the detector temperature at a known reference is a major requirement, which is either realized by cooling systems or by recording the detector temperature. Furthermore, the radiation reaching the detector is contaminated by radiation emitted by the imager (body, lens) itself. As it is not always possible that the sensor temperature is stabilized by a cooling system or that the influence of*

*the instruments temperature on the measurements can be well quantified, many TIR imagers apply on-board calibrations with black bodies."*

**l.69: when presenting a new instrument, it's useful to point how it differs from existing available (sometimes commercially) instruments, here or later on in the manuscript.**

Yes, so far we just mentioned a couple of instruments but did not put VELOX into context with them. We have now put more efforts into this and have included a comparison to other instruments like the Mineral and Gas Identifier (MAGI; Hall et al., 2015) and MAKO (Hall et al., 2016), and the airborne thermal-infrared hyperspectral imaging system (ATHIS; Liu et al., 2020).

*"Notable instruments are the enhanced MODIS Airborne Simulator (EMAS-HS; Guerin et al., 2011), or the far infrared radiometer (FIRR; Libois et al., 2016), which helped to develop cloud and surface retrieval products based on TIR observations. Advanced airborne pushbroom TIR imagers are for example the high performance Mineral and Gas Identifier (MAGI; Hall et al., 2015) and MAKO (Hall et al., 2016) and the Airborne Thermal-infrared Hyperspectral Imaging System (ATHIS; Liu et al., 2020), which provide up to 128 spectral channels on up to 2800 spatial pixels."*

*"Here we describe the temperature-stabilized TIR imager VELOX (Video airbornE Longwave Observations within siX channels). VELOX does not apply an on-board calibration, which is compensated by operating the imager in a temperature stabilized housing and by a series of post-calibration routines. This reduces the size of the imager setup, an advantage that distinguishes VELOX from most of the known airborne TIR imagers. The imaging sensor of VELOX is commercially available and manufactured by the IRCAM GmbH, Erlangen, Germany. It measures radiance in six spectral bands in the TIR wavelength range from 7.7 µm to 12.0 µm, which fits well in the wavelength range that is commonly chosen for TIR measurements and suits common cloud and surface retrieval. Although the number of spectral channels is lower compared to the many of the airborne TIR imagers introduced above, VELOX provides two-dimensional (2D) images with high temporal resolution. Compared to common line scanners (e.g., MAS, MAGI, or MAKO) this allows a wider range of applications, like analysing the horizontal fine-scale structure of clouds and dynamic processes at clouds edges or stereoscopic image processing utilizing the large overlap of the individual images. VELOX is currently implemented into the remote sensing configuration of HALO (Stevens et al. 2019) for airborne observations, but can potentially be operated on other airborne platforms and as a ground-based sky imager similar to Schäfer et al. (2013) or Jäkel et al. (2013). The integration of VELOX makes the HALO cloud-observatory instrumentation fully analogous to the sensor package being flown on the EarthCARE satellite, combining active radar and lidar observations with passive solar, thermal infrared and microwave remote sensing."*

*"J. L. Hall et al., MAGI: A New High-Performance Airborne Thermal-Infrared Imaging Spectrometer for Earth Science Applications, in IEEE Transactions on Geoscience and Remote Sensing, 53, 10, pp. 5447-5457, 2015, doi: 10.1109/TGRS.2015.2422817, 2015."*

*"J. L. Hall et al., Mako airborne thermal infrared imaging spectrometer: performance update, Proc. SPIE 9976, Imaging Spectrometry XXI, 997604, https://doi.org/10.1117/12.223924, 2016."*

*"C. Liu et al., "New Airborne Thermal-Infrared Hyperspectral Imager System: Initial Validation," in IEEE Journal of Selected Topics in Applied Earth Observations and Remote Sensing, 13, 4149-4165, doi:10.1109/JSTARS.2020.3010092, 2020."*

**l.73: repetition of reference for EarthCARE**

Deleted

**l.93: what's the size of the filter wheel? Of the filters? As they do not appear in Fig. 1 I assume they're quite small**

The size of the filter wheel is approximately smaller than 10 cm in diameter. Each filter has a diameter of 25 mm and a thickness of 1 mm. We've added the information of the filter size in the text.

*"A synchronously rotating filter wheel, providing six slots for spectral filters (each 25 mm in diameter), is mounted between the lens and the detector."*

**l.94: 100 Hz is for a complete rotation or to go from one filter position to the next? Clarify the link with the 100 Hz acquisition. A single frame on each filter or integration of multiple frames on the same filter? State here that the full measurement on all filters takes 0.06 s, this is a major information.**

It's true, this is important information. We revised this part accordingly.

*"… Therefore, the whole filter wheel rotates with a frequency of 100 Hz. With six channels, this results in a frame rate of about 15 Hz, meaning that a full filter-wheel spin to separately acquire six single images (one with each of the six filters) takes 0.06 sec. …"*

**l.95: *partly adapted* is unclear. Are the filters meant to match MODIS filters characteristics or only the central wavelength?**

We've changed it to the following:

*"…, which are adapted to the channels (centre wavelength) popular among satellite instruments, …"*

**Table 1: what is the temperature reference for the NETD? Could you provide more details in the text about the cooling of the sensor (temperature, stability etc.)**

For the stand-alone thermal imager (without filters) the manufacturer provides a NETD value of ≤ 40 mK (typically 30 mK @ 25°C). In Sect. 3.3 we recalculate this value taking the installed filters into account. The manufacturer of the infrared thermometer provides a typical value of 100 mK, which is temperature dependent. We've added the information for the stand-alone instruments in Table 1 and further included some information on the cooling of the sensor in the text.

*"The camera system installed in VELOX comprises two components: an actively cooled TIR imager (VELOX327k veL), and an un-cooled infrared thermometer (Heitronics KT19.85II), which serves as a secondary reference."*

New paragraph added:
*"The 2D detector is actively temperature controlled and cooled to 65 K by a Stirling cooler. This helps to minimize the influence of the environmental conditions on the performance of the imager, which ensures a wide temperature range for the measurements and a stable absolute calibration. The active*

*cooling reduces the Noise Equivalent Differential Temperature (NEDT) to 40 mK (without filter) and allows for the observation of small temperature differences, in this case of ocean and sea-ice surfaces as well as clouds, with a target temperature between 233.15 K (-40°C) and 373.15 K (100°C)."*

**l.102: is the spectral response of the detector really zero outside of the range 7.7 – 12 microns?**

The specification for the Mercury Cadmium Telluride (MCT) detector are provided by the manufacturer. We cannot rule out, that the spectral response will immediately decrease to zero outside the 7.7 μm and 12 μm range. However, as the correction of the radiation internally emitted and reflected by the filter works properly, when accounting for emission/reflection between 7.7 μm and 12 μm, there is no indication that a significant amount of radiation outside the specified range is detected by the sensor.

**l.102: maybe state here the reasons (if any) for duplicating this broadband channel**

Indeed, the reasons for duplicating both channels were not clear. Up to now, we just need the four narrow-band channels and one broadband channel. However, the filter wheel is only provided with six slots. To ensure the same optical path with respect to the four narrow-band filters (equally sharp images), an optically transparent window needed to be installed for the broadband channel. In addition, to ensure a stable rotation of the filter wheel, the remaining sixth slot (located within the filter wheel on the opposite side of the broadband channel) needed to be filled as well. We could have also chosen a fifth narrow-band filter for the sixth slot, but we wanted to wait until we gain more experience with the operation of the filter and their limitations (low signal, minimum width, etc.) before choosing an additional filter. To avoid any confusion, we have included the information on why we use optically transparent windows instead of just leaving it empty and why we've used the same window two times.

*"… For some applications, broadband measurements, which provide a higher accuracy are sufficient, e.g., for sea surface temperature or cloud top altitude retrievals. Therefore, two of the filter slots provide redundant broadband measurements (7.7 - 12 μm, Channel 1 and 4 in Tab. 2), but are fitted with an optically transparent window to match the optical paths of the other channels."*

**l.144: I'm not sure to get what the first issue is. Is it to project the pixels at the Earth surface? Is the aircraft movement used to tackle this, or just the position (including attitude)? Does it work when the emitter is not the surface, but a cloud?**

We guess the wording "position" and "aircraft movement" were misleading. We revised the sentence to make it clear that we talk about the pixel orientation (sensor zenith and azimuth angles) including the aircraft attitude. The estimation of the pixel size is surface related, but would work as well for clouds, if their top altitude is known. In the revised version we included information on the pixel size.

*"The geometric characterization of the VELOX system addresses two main issues: (i) the relative pixel orientation (sensor zenith and azimuth angles) and size (ground or cloud top related) as a function of aircraft attitude and altitude, …"*

In Sect. 3.1.1 Viewing Geometry
*"The rectilinear ground pixel size is estimated by trigonometric relations from the sensor zenith angles and the aircraft altitude. In case of clouds and a known cloud top altitude, horizontal pixel size at cloud top can be estimated."*

**l.169: can the difference in acquisition time for different filters be an issue as well? To be related with the distance travelled by the aircraft between two successive filters**

The relevant paragraph in the manuscript tackles the image shift due to geometric inhomogeneities of the filter surface (in a non-moving system). Integration time or aircraft movement doesn't matter in this regard.

The influence of different integration times for different filters can be neglected. With integration times of 50 µs or 70 µs and an average flight speed of 220 ms$^{-1}$ (for HALO), the aircraft is just moving by 11 mm to 15 mm.

Off course, the distance travelled by the aircraft between two or more successive filters will lead to a noticeable shift of the acquired scene. With 100 Hz frame rate and 220 ms$^{-1}$ travel speed the scene acquired after a full rotation of the filter wheel is shifted by approximately 13 m. The scene between two successive filters is shifted by approximately 2 m.

However, we see the point. From the current title, the reader might expect an explanation on how to handle scene shifts between the single channels due to the aircraft movement. Therefore, we've changed the title of this section.

Because we think that only the shifts due to inhomogeneities of the filter surfaces are an issue for the calibration and the shifts due to aircraft movement are rather a data post-processing issue, we focus in this section on the filter issue only. Information on how to handle shifts due the aircraft movement are included in Sect. 4.1, where we also show the brightness temperature images of the same scene acquired at different times.

New headline Sect. 3.1.2
*"Image shift due to filter geometry"*

Added in Sect. 4.1
*"To combine different spectral channels, the successive images need to match pixel wise. This requires to account for the horizontal shift between the images. This shift is estimated from the aircraft speed and the projected pixel size, while the latter depends on the distance between the surface/cloud and the aircraft (see Sect. 3.1)."*

**l.175: this scene identification/matching deserves more details. What kind of algorithm is used?**

Actually, in that case the identification/matching was done manually by comparing six images acquired with the different filters and testing it for several different scenes. As these offsets are stable, we think, this manual approach is sufficient. We've added "manually" to the sentence.

*"… After recording a test image (e.g., chess board like) for each channel, all parts of the scene are that are covered by each image are identified manually. …"*

**l.189: In a system without on-board calibration, this calibration procedure is crucial. Can you provide more details on the way the corrections were obtained (lab experiments to isolate the impact of temperature changes?). At least consider referring to a paper detailing how this is achieved.**

The corrections mentioned in line 189 are part of internal calibration/correction procedures, which are provided in the software kid of the manufacturer. The corrections are based on an *advanced internal radiometric calibration model*, which makes use of several internal parameters but is not published in full detail. Therefore, we limit our description of the corrections to the basic concept. For example, a changing body temperature of the imager or variations in the filter wheel temperature are considered when the absolute radiometric calibration is applied to the raw data to calculate the brightness temperature or radiance. However, to avoid any confusion we revised the relevant part in the

manuscript a bit to make this clearer. Furthermore, we added additional information on this topic in Sect. 3.2.3: Radiometric cross-calibration for field operation (please see our answers to your comment on line 214).

*"The TIR imager is provided with an advanced internal radiometric calibration model, developed and validated by the manufacturer over many years (Schreer et al. 2004). This model includes calibration factors to transfer the raw counts into brightness temperature or radiance and takes changes in the imagers body, optics, and filter temperatures into account."*

**l.205: it is not clear what the link between *gain/offset* and *variable ground potentials* is. Once the non-uniformity of the pixels is identified, how is the amplitude of the correction determined? Why are stripes removed with this procedure? Do these stripes come from pixels with different gains or different offsets (due to straylight for instance)?**

We've revised this paragraph to better combine the information related to gain/offset and variable ground potential.

The amplitude of the correction is defined by the absolute radiometric calibration, which is fitted to the slope and number of counts that originate from the difference of the number of counts per integration time (or target temperature) for the two settings (low/high integration time or target temperature). The fit and the absolute radiometric calibration function are both different for each spatial pixel, which results in a different non-uniform correction for each spatial pixel. However, by fitting the absolute radiometric calibration function to the slope and range of counts, which are given by the measurements with the two settings, the response of all spatial pixels is homogenized. This removes noise and stripes from the image. The noise and stripes are related to different gains of the single spatial pixels, which are more or less fixed. However, due to the variable ground potential of the spatial pixels, which changes after each restart of the imager, no fixed correction can be applied; the non-uniform correction always needs to be newly generated for each restart of the instrument.

*"… Furthermore, each spatial pixel has a slightly different gain and its ground potential has no fixed value. The latter changes slightly after each restart of the imager. Both the different gain and the variable ground potential naturally imprint noise into the measured images. To remove these effects, a two-point non-uniform correction (Budzier and Gerlach, 2015} was applied prior to each flight, which homogenizes the response of all single detector pixels among each other.*
*The non-uniform correction is performed at the ground and generated after the sensor chip is cooled down to its stable operating temperature. It requires at least two measurements of a homogeneous target (e.g., non-reflecting plate or black body) providing different intensities to determine the gain of each spatial pixel. Afterwards, by fitting the pixel-dependent absolute radiometric calibration function to this pixel-wise gain, the response functions of all spatial pixels are homogenized among each other, which removes noise and stripes from the image.*
*The different intensities provided by the target can be realized either by a black body set to two different temperatures or by using two different integration times, which artificially change the intensity recorded by the sensor. During EUREC[4]A, …"*

**l.207: does the scene need to be homogeneous to apply this calibration, as stated above?**

Yes, this is necessary. Otherwise the inhomogeneous structure of the target used for this calibration would be imprinted in the images. Please also see our answer on your comment to l. 205.

**l.214: I don't understand why this specific calibration is not implemented directly at the step 3.2.1. Practically, is the correction pixel-dependent? Is it static or does it depend on environmental conditions?**

You are right, this step could be also implemented directly in Sect. 3.2.1: "Internal radiometric calibration model". However, we decided to separate it from this step, because Sect. 3.2.1 deals with the basic calibration model provided by the manufacturer, which is fixed and does not account for environmental changes of the camera system. All calibration steps, which need to be repeatedly adapted to the measurement environment after each start of the camera are discussed in Sect. 3.2.3: "Radiometric cross-calibration for field operation".

The correction is not pixel dependent. It is more about general offsets in the measured brightness temperature or radiance related to the installation of the imager in the pressurized tube, which provides different environmental conditions compared to the initial radiometric calibration performed by the manufacturer in the lab.

A further reason for the additional cross-calibration is to adapt to changes in the radiometric calibration that might occur as a result of the instrument aging. We've revised the relevant part in the manuscript as follows:

*"The internal absolute radiometric calibration of the TIR imager is fixed unless no new radiometric calibration is performed in the labs of the manufacturer. It cannot be adapted during field operations and, therefore, it cannot account for changes of the system performance in a different environment. Only absolute offsets in the measured signal can be corrected for. To monitor the stability of the radiometric calibration with respect to the instrument aging and to adapt it to the environmental conditions (installation in the pressurized tube) during the research flights, cross-calibrations were performed with a mobile black body unit. In the laboratory, …"*

*"Knowing the conditions during the flights and estimating the related offsets in advance, this step can be also directly combined with the internal radiometric calibration model introduced in Sect. 3.2.1."*

**Eq. 1: how where the different parameters of this equation determined? Was the method validated by cross-calibration against a black-body?**

The parameters were obtained from the manufactures of the window and the lens and validated by cross-calibrations with a black body. We've added some more information on how to handle the equation and from where we got the values. In this regard, we also included a plot of the spectral transmission, reflection, and absorption/emission coefficients and had to reorganize some paragraphs to improve the readability. The new version reads like this:

*"… The specific absorption/emission is given by $\varepsilon$, the reflection by R, and the transmission by T. The subscripts denote the two coefficients of the window (win) and the imager's lens (lens). The spectral $\varepsilon$, R, and T of the window as provided by the manufacturer and validated by cross-calibrations with a black body are shown in Fig. 3. Overall, the Germanium window has a high average transmissivity of 93.95 % in the wavelength range from 7.7 µm to 12 µm. The spectral behavior of the reflection coefficient is rather constant over the entire range with about 5 % on average, while the absorption/emission coefficient is almost negligible for the VELOX channels 2 and 3, but affects longer wavelengths (up to 10 % for Channel 5 and 6). The emission coefficient of the lens is 0.15. Although, this value seems to be quite large, it results in a rather low contribution to the composed signal ($\approx 0.75$ %), because it only corresponds to the radiation emitted by the lens. For the application of Eq. 1 the window parameters were integrated for the filter response function of the selected spectral channel.…"*

[Figure]

*Figure 3: Spectral transmission, reflection, and absorption/emission coefficients of the Germanium window for the wavelength range covered by the TIR imager. Included are in addition the response functions (transmission coefficients, dashed/dotted lines) of the four narrow-band channels.*

**l.268: what is *accuracy* here? Absolute accuracy detailed just below?**

No, it is not the absolute accuracy detailed in the next paragraph. However, we agree that the wording was misleading. We intended to say that a high NETD doesn't necessarily lead to useless images. If the contrast in the images is large enough, then the thermal signal is still useful to resolve spatial structures of the target. We revised it accordingly:

*"However, as long as the image values provide a significant contrast, the acquired thermal signal allows to resolve spatial structures of the target."*

**l.297: how can you know that no cloud-free ocean was observed?**

The highest brightness temperatures observed along the flight track are attributed to the cloud-free regions as we can assume that the highest temperatures are related to the warm ocean surface. Changes of the brightness temperature of the ocean surface are expected to be small in comparison to sudden temperature drops induced by clouds. Therefore, if the highest brightness temperature observed within a 60-second sequence is reduced by more than 3 % compared to the previous 60-second sequence, it is highly likely that clouds were present within this time frame. In this case, the calculated maximum brightness temperature envelope is set to the value of the previous cloud-free sequence. Using the 2D images, this method was visually validated for different cloud situations. We revised the part and added more information:

*"… If a 60-second sequence is fully covered by clouds, the maximum values of the previous cloud-free sequence is used for the envelope. This is justified, because temperature changes of the ocean surface can be assumed to be spatially (temporally) weaker compared to the effect of clouds. A 60-second sequence is defined fully cloudy, if its maximum brightness temperature is reduced by more than 3 % compared to the previous sequence. ..."*

**l.300: are the differences between simulated and estimated cloud-free BT due to differences in atmospheric state, or could they be solely explained by measurement uncertainty? The differences should be compared to measurement uncertainty on the one hand, and to simulation variability on the other hand. Are the points away from the 1:1 line actually those acquired far from a dropsonde?**

You are right. Here, we've certainly missed some explanation in the text. We performed an analyses following your proposal. To investigate if the larger differences are related to the distance to the next dropsonde, we've tested it once with all data and once with measurements acquired during the circular

flight sections only. Within the circular flight sections, we usually had dropsonde launches approximately every 5 min. In comparison, during the flight sections to ships or buoys only one or two dropsondes within one hour were released. The differences are illustrated in Fig. R1 below. The largest differences can be observed for measurements that were performed far away from the next dropsonde (during excursions). In comparison, the measurement uncertainties are a minor reason for these differences. This is indicated by the very similar spread of the differences derived from the different channels, although they have very different NETD ranging from 48 mK to 605 mK.

[Figure]

Figure R1. Correlation between the cloud-free simulations and the maximum brightness temperature values displayed in Fig. 6, which represent cloud-free measurements over the warm ocean. Left: old version, right: new version.

Therefore, both investigations strongly indicate that the main issue is related to the dropsonde distance and its influence on the simulation. During the flight sections in between two dropsondes, the simulations are assumed to be constant (only influenced by flight altitude, which is quite stable), while the atmosphere continuously changes (please also compare the differences of the temporal resolution in the time series for sections with high dropsonde frequency (white area) and low frequency (grey shaded area) in Fig. R2).

[Figure]

Figure R2. Time series of brightness temperature (black, Channel 1) observed by the central 10 by 10 pixels of the thermal imager and the corresponding cloud-free simulations (red) during the EUREC[4]A flight on 13 February 2020. Grey shaded areas mark flight sections with low dropsonde density.

In the original version we have shown only the circular flight patterns and excluded the flight sections to the ships without explaining it. To make the point clearer we've exchanged it with the plot that includes all data now. Furthermore, we elaborated the explanation.

*"The correlation R between the maximum envelope fit and the simulations is 0.84 (Pearson correlation). Tests have shown that the measurement uncertainties have a minor influence on these differences. Although the NETD varies significantly between the single channels, the spread of the deviations observed with the different channels shows a similar pattern. The reason for the observed differences is mainly linked to the spatial/temporal resolution of the simulations, which are limited by the*

*frequency of dropsonde releases (between one sonde per 5 min and per 1h). Within such a period, the simulations remain constant, while in reality, the atmosphere might change continuously. Excluding flight sections with low dropsonde density (e.g., 11:30 UTC to 13:30 UTC in Fig. 6) removes the largest differences, which supports this assumption. However, …"*

**l.317: what is the interest of such a comparison with pushbroom configuration, if the obtained differences are not better described?**

The word "comparison" was probably misleading. The paragraph is rather meant to describe how the data can be "combined" with traditional pushbroom imagers or nadir-pointing instruments. To further promote the benefit of processing the data in such way, we've also added more ideas for possible applications.

*"For combining with traditional pushbroom imagers …"*

*"Further applications for such a combination between VELOX and, e.g., specMACS are cloud retrievals that are based on measurements in the solar and thermal wavelength range and the investigation of the differential surface warming due to simultaneous cloud shadowing and cloud base emission."*

**l.347: for this, is the average of 10x10 pixels used or it is performed for individual pixels? Does the maximum envelope come from the time series of individual pixels or from a single image?**

Yes, the maximum envelope approach results from the time series of the averaged brightness temperature of the central 10 by 10 spatial pixels. To avoid any confusion, we included more information on that.

*"The maximum envelope as derived from the central 10 by 10 spatial pixels, served as a reference in the cloud mask algorithm described above. Assuming that the envelope value is valid for the full 2D VELOX images, a 2D cloud mask and cloud fraction for each image was derived."*

**l.368: here it is somehow assumed that the cloud is optically thick and that the emission comes from the top of the cloud. Can you discuss a bit these assumptions and their limits? Would changes in LWC or $r_{eff}$ make a difference on the emissivity fixed to 0.99?**

You are right. Applying this assumption, optically thick clouds are automatically assumed. Geometrically and optically thinner clouds with a lower liquid water content would result in lower emission coefficients, while the influence of the effective radius on the emissivity would become important for wavelength larger than 11.5 µm. However, changes due to small variations in the emissivity are of minor importance considering the other uncertainties introduced by matching with the dropsondes. Therefore, we think that for a first approximation it is sufficient to use a fixed coefficient for the emissivity.

However, related to your next comment on line 376, we decided to skip the method, which is based on VELOX and dropsonde data only. We think it gives more a first estimation of the cloud top altitudes than providing a validated result. Instead, we cross-calibrate the VELOX data directly by measurements of WALES. The purpose of this method is to extent the point measurements by WALES along the flight track to 2D maps of cloud top altitudes. Doing so, it is not necessary to assume an emission coefficient. The revisions for the whole Sect. 4.3 are listed below your comment referring to line 376.

**l.376: 470 m offset seems huge for a cloud mostly ranging from 600 to 1400 m. Can it really be explained by errors in actual atmospheric profile? How does an error in BT translate into an error in cloud top altitude, roughly (for the atmospheric profiles observed)?**

These offsets are randomly distributed (see. Fig R1a). The mean of the offset distribution is located close to 0 m for all flights. The 25 % percentile is about 120 m. For a flight leg with low dropsonde coverage (see Fig. R1c), 50 % of the VELOX cloud top altitudes had an offset smaller than 400 m (for low temporal resolution of 1 Hz) in pointwise comparisons with WALES. This offset could be further reduced by the use of a temporally higher resolved time series (full VELOX resolution, Fig. R1b).
This improvement indicates that the offsets are affected by the differences between the VELOX and WALES cloud masks, especially due to mismatches at cloud edges. A second source for larger offsets is attributed to the distance to the closest dropsonde, which introduces uncertainties in the radiative transfer simulations and the parametrization of the cloud-free atmospheric temperature offset. This is confirmed by the analysis of flight sections with a low number of dropsonde launches (2 February 2020). In these cases, the offsets increase as illustrated in Fig. R1c.

[Figure]

For EUREC[4]A flight on 2 February 2020, **(a)** offsets beween cloud top altitude from VELOX and WALES, **(b)** Absolute offsets beween cloud top altitude from VELOX and WALES compared for 1 Hz VELOX data (red) and full VELOX temporal resolution (purple), and **(c)** offsets for flight sections with low dropsonde density. Red/purple lines indicate median values.

The revised section reads now like this:

*"The cloud top temperature measured by VELOX is closely linked to the cloud top altitude. This relation is commonly used in cloud top altitude retrievals from satellite observations. Here, a similar approach is used for the images from VELOX and extended by a cross-calibration with nadir-pointing cloud top altitude measurements from WALES (Wirth et al., 2021). This method allows to extend the nadir measurements of WALES to 2D maps of cloud top altitudes, which resolve the horizontally structure of shallow cumulus. To apply the cross-calibration, a first guess of cloud top altitude from VELOX is needed. It is derived from the measured brightness temperature of the thermal imager's broadband channel 1. This first guess is necessary since there is no fixed direct relation between cloud top altitude derived from WALES and the VELOX brightness temperature along the flight path. It rather varies in time with the changing influence of the atmosphere. For the first guess, the brightness temperature is combined with atmospheric profiles from dropsondes (George et al., 2021) and radiative transfer simulations of the cloud-free atmosphere. The simulated brightness temperatures are parametrized as a function of the distance to the cloud top and used to invert the measurements.*
*In a second step this first guess of the VELOX cloud top altitude is cross-calibrated with the WALES cloud top altitude. The cross-calibration uses the cloud mask ("most-likely-cloudy" threshold) of VELOX (cloud mask based on the central 10 by 10 spatial pixels) and WALES. If both instruments detect a cloud, the*

*cross-calibration is applied, which links the first guess of the VELOX cloud top altitude to the WALES cloud top altitude in a linear relationship. At this juncture, the correction of the first guess VELOX cloud top altitude ranges between 100 m and 300 m.*

*Two major reasons for these uncertainties were identified; (i) an increased distance to the next dropsonde leads to uncertainties in the cloud-free simulations, and (ii) missmatches in the cloud mask by VELOX and WALES. The latter can be reduced when using the full temporal resolution of VELOX.*

*Considering the NETD of VELOX, the full approach allows a retrieval of 2D maps of cloud top altitudes with a vertical resolution of 40 m. As an example, Fig. 11c shows the derived cloud top altitude for the cloud scene from 9 February 2020. Cloud top altitudes below 600 m might be nonphysical and are related to very thin clouds or cloud edges. These low cloud top altitudes probably results from a contamination of the signal by the emission of the ocean below."*

[Figure]

Figure 11. (a) Two-dimensional field of brightness temperature measured at a flight altitude of approximately 10 km with the VELOX broadband channel between 7.7 µm and 12 µm during the EUREC4A field campaign on 9 February 2020 at 15:05:21 UTC. For the same scene, panel (b) shows the combined cloud mask and panel (c) the retrieved cloud top altitude.

**l.378: can you detail this correction procedure since it may be critical (when errors are larger than the measured range of variations)**

Yes, we did. Please see our reply to your comment to line 376.

**l.396: references to EUREC4A not needed here**

Removed

**l.408: typo: *Oceanc***

Corrected to *"Ocean"*

**l.433: how is set ocean emissivity?**

Thanks for pointing out that we've missed a description of the origin of the ocean emissivity ($\varepsilon_{ocean}$). It is derived by subtracting a wavelength dependent ocean albedo ($\alpha_{ocean}$) file from unity ($\alpha_{ocean} = 1 - \alpha_{ocean}$). The applied ocean albedo file is provided by the International Geosphere Biosphere Programme (IGBP; Belward and Loveland, 1996)

*"The ocean emissivity ($\varepsilon_{ocean}$) is derived by subtracting a wavelength dependent ocean albedo ($\alpha_{ocean}$) file from unity ($\varepsilon_{ocean} = 1 - \alpha_{ocean}$). The applied ocean-albedo file is provided by the International Geosphere Biosphere Programme (IGBP; Belward and Loveland, 1996)"*

*"Belward, A. and Loveland, T.: The DIS 1-km land cover data set, GLBAL CHANGE, The IGBP Newsletter, 27, 1996. "*